# Homogenizing interfacial assembly via indole-mediated binary monolayers for perovskite solar cells

Haojiang Shen[1,10], Yeming Jin[1,10], Fuqiang Li[1,10], Kangning Zhao [2,10], Nan Shen[1 ✉], Jie Yang[1], Ya Li[1], Zhengyuan Long[1], Yuxuan Sheng[1], Hongbing Li[3], Fei Guo[3], Zong-Xiang Xu [4], Yong Ding [5], Xingzhu Wang[1,6 ✉], Geping Qu[7 ✉], Shi Chen [1 ✉] & Mohammad Khaja Nazeeruddin[8,9 ✉]

Self-assembled monolayers (SAMs) have emerged as efficient hole-transport layers for inverted perovskite solar cells (PSCs), yet molecular self-aggregation during assembly limits interfacial homogeneity and device performance. Here we report an indole-carbazole co-adsorption strategy by incorporating N-indoleacetic acid (Nd) into (4-(3,6-diphenyl-9H-carbazol-9-yl)butyl)phosphonic acid (Ph-4PACz) to construct phase-homogeneous monolayers. Nd interacts with Ph-4PACz via synergistic π−π stacking and hydrogen bonding, resulting in a uniform alternating Ph-4PACz/Nd molecular arrangement. This co-adsorbed structure enables optimized interfacial energy alignment, enhances perovskite film uniformity, and suppresses trap-assisted non-radiative recombination. As a result, devices achieve an efficiency of 26.95% (certified 26.57%) on $0.0717 \, cm^2$ and 25.61% on $1 \, cm^2$, retaining 93.36% of their initial efficiency after 1500 h of maximum power point tracking under continuous illumination and 91.10% after 1200 h at 85 °C. The strategy is broadly applicable to carbazole-based SAMs and wide-bandgap PSCs, offering a general co-adsorption route toward efficient and stable devices.

Self-assembled monolayers (SAMs), such as (2-(9H-carbazol-9-yl)ethyl) phosphonic acid (2PACz) and its derivatives, have emerged as promising hole transport layers (HTLs) in inverted perovskite solar cells (PSCs)[1–4]. These SAMs typically consist of anchoring groups, spacer units, and conjugated functional groups (Fig. 1a), offering favorable energy-level alignment, reduced interfacial trap density, and efficient hole extraction[5]. In particular, anchoring groups such as phosphonic

acid or carboxylic acid can chemically bond with hydroxyl groups on indium tin oxide (ITO) or fluorine-doped tin oxide (FTO), enabling the formation of ultrathin and conformal monolayers at the anode interface[6,7]. However, the presence of highly polar groups often induces uncontrolled intermolecular interactions during the self-assembly process[8,9]. These non-directional interactions can lead to local molecular aggregation or the formation of nanoscale voids within the SAM

[1]State Key Laboratory of Green Chemical Synthesis and Conversion, School of Energy Science and Technology, Henan Key Laboratory of Quantum Materials and Quantum Energy, School of Future Technology, Henan University, Zhengzhou, China. [2]School of Physical Science, Great Bay University, Dongguan, China. [3]Institute of New Energy Technology, College of Physics and Optoelectronic Engineering, Jinan University, Guangzhou, China. [4]Department of Chemistry, Southern University of Science and Technology, Shenzhen, Guangdong, China. [5]School of Renewable Energy, Hohai University, Nanjing, China. [6]Engineering and Research Center for Integrated New Energy Photovoltaics & Energy Storage Systems of Hunan Province, School of Electrical Engineering, University of South China, Hengyang, China. [7]Department of Materials Science and Engineering, City University of Hong Kong, Kowloon, Hong Kong, China. [8]School of Integrated Circuits, Southeast University, Wuxi, Jiangsu, China. [9]Institut des Sciences et Ingénierie Chimiques, Ecole Polytechnique Fédérale de Lausanne (EPFL), Lausanne, Switzerland. [10]These authors contributed equally: Haojiang Shen, Yeming Jin, Fuqiang Li, Kangning Zhao. ✉e-mail: shennan@henu.edu.cn; xingwang0926@163.com; gepingqu@cityu.edu.hk; chenshi@henu.edu.cn; mdkhaja.nazeeruddin@epfl.ch

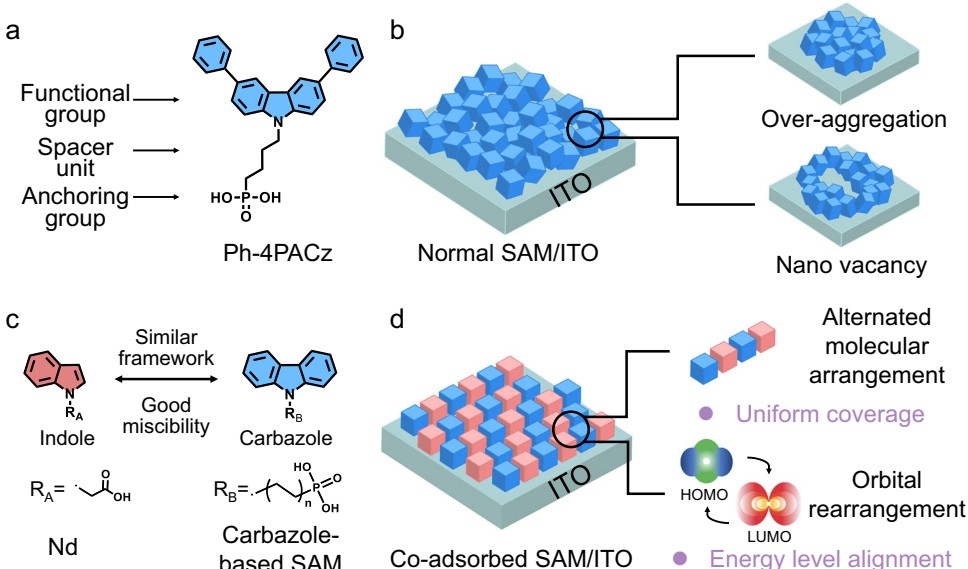

**Fig. 1 | Design of co-adsorbed SAMs. a** Chemical structure of Ph-4PACz.
**b** Schematic illustration of normal SAM deposited on ITO. **c** Chemical structures of
Nd and carbazole-based SAMs, as well as their structural similarity. **d** Schematic
diagram of alternating co-absorbed SAM deposited on ITO.

(Fig. 1b), thereby compromising interfacial uniformity and hindering
efficient charge transport across the buried interface[10–13].

Several strategies have been developed to improve SAM coverage
and interfacial uniformity, including deposition process optimization,
bilayer design, and additive engineering. For instance, vacuum-
deposited (4-(3,6-dimethyl-9H-carbazol-9-yl)butyl)phosphonic acid
(Me-4PACz) improved surface wettability and promoted better per-
ovskite film morphology without significantly altering interfacial
chemistry[14]. Bilayer architectures combining $NiO_x$ and SAMs demon-
strated enhanced coverage and reduced interfacial losses, while small-
molecule additives such as methyl phosphonic acid (MPA) and
3-mercaptopropionic acid (3-MPA) helped suppress aggregation and
minimize interfacial energy losses[15,16]. Despite these advances,
achieving molecular-level homogeneity and indeed long-range order
within SAMs remains a daunting challenge. In addition, the debate
surrounding the impact of ordered versus disordered SAMs on charge
transport and device stability necessitates further elucidation[17–19]. Most
existing strategies focus on macroscopic-level modifications and
provide limited control over molecular-scale phase behavior and
structural miscibility. Alloy-like molecular architectures have recently
emerged as an effective strategy for regulating the morphology and
electronic properties of organic semiconductors[20–22]. This approach
relies on the homogeneous co-assembly of structurally analogous yet
non-identical molecules with high mutual miscibility[23,24]. Such mole-
cular alloying enables directional control over film formation dynam-
ics, effectively suppressing aggregation and void formation while
promoting the growth of uniform and compact layers. Beyond inte-
grating the individual advantages of each molecular component, this
strategy can also induce emergent properties, including orbital-level
modulation, enhanced charge transport channels, and optimized
interfacial energetics[25,26].

Inspired by this concept, we developed an indole-carbazole co-
adsorbed SAM strategy to effectively suppress undesirable molecular
aggregation and enable phase homogeneity by co-assembling N-
indoleacetic acid (Nd) with (4-(3,6-diphenyl-9H-carbazol-9-yl)butyl)
phosphonic acid (Ph-4PACz). Theoretical simulations and experi-
mental characterizations revealed that the preferential binding of Nd
with Ph-4PACz via enhanced π-π stacking and hydrogen bonding
induced the formation of highly uniform, alternating Ph-4PACz/Nd
molecular arrangement in the co-adsorbed SAM. Meanwhile, such

architecture with nonlinear energy levels enabled favorable band
alignment with the perovskite, and also enhanced perovskite homo-
geneity, leading to faster charge transfer and slower defect-assisted
non-radiative recombination. Profiting from phase homogenization,
single-junction devices adopting the alternating co-adsorbed SAM
achieved a champion power conversion efficiency (PCE) of 26.95%
(certified 26.57%) on 0.0717 $cm^2$, together with a PCE of 25.61% over
1 $cm^2$. These devices exhibited impressive operational and thermal
stabilities, maintaining 93.36% and 91.10% of their initial PCEs after
1500 h of maximum power point tracking under illumination and
1200 h heating at 85 °C, respectively. Importantly, this strategy is
broadly applicable to a wide range of carbazole-based SAMs and wide-
gap PSCs, offering a generalizable platform for molecular interface
engineering in efficient and stable PSCs.

## Results
### Design of co-adsorbed SAMs
As shown in Fig. 1a, c, Ph-4PACz and Nd, based on a fused pyrrole-
benzene framework, possess carbazole and indole cores, respectively.
This structural similarity between Ph-4PACz and Nd enables compar-
able electron delocalization and provides a molecular basis for their
miscibility. The interaction parameter (χ) analysis based on contact
angle measurements confirms sufficient miscibility between Ph-4PACz
and Nd (Supplementary Fig. 1 and Supplementary Table 1), where the χ
between Nd and Ph-4PACz is close to that between Ph-4PACz and
2PACz, potentially benefitting the formation of homogeneous SAMs[27].
To investigate the evolution of their intermolecular aggregation
behavior, ultraviolet-visible (UV-Vis) absorption spectra were collected
for Ph-4PACz films mixed without and with Nd (denoted as Ph-Nd).
Notably, decreasing the Ph-4PACz:Nd mixture ratio from 1:0 to 1:2
induced a progressive blue shift from 297 to 295 nm in the main
absorbance peak of the Ph-4PACz film (Supplementary Fig. 2a).
Moreover, an obvious photoluminescence (PL) quenching was detec-
ted for the mixed Ph-Nd solution compared with pure Ph-4PACz
solution (Supplementary Fig. 2b). These results indicate the intro-
duction of Nd leads to a transition from a disordered aggregation in
Ph-4PACz to an ordered face-to-face aggregation (H-type) in the Ph-Nd
binary system[28,29].

To further examine the electronic structure of Ph-4PACz films
without and with Nd, we performed cyclic voltammetry (CV)

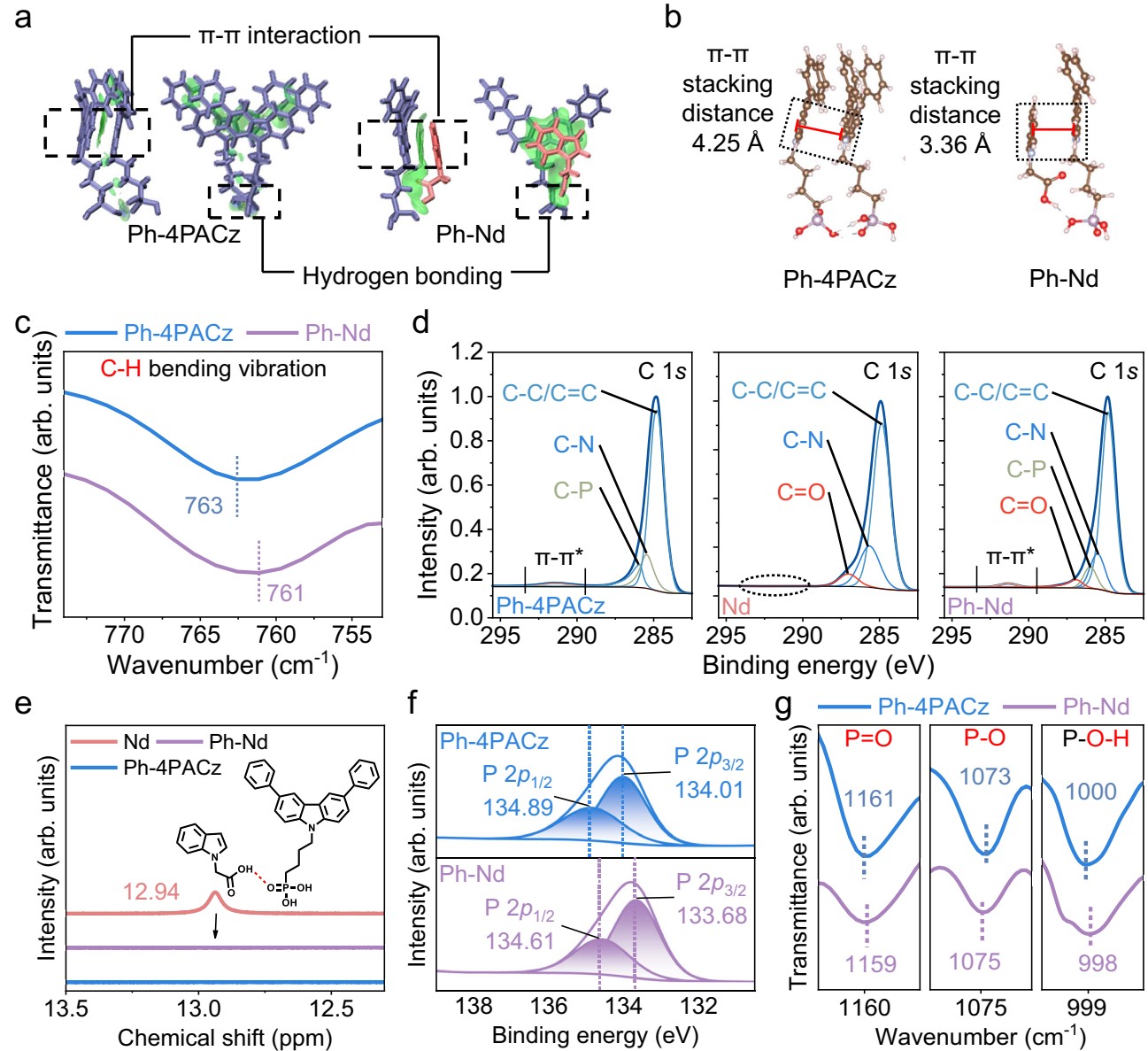

**Fig. 2 | Interactions within the co-adsorbed SAMs. a** IGMH analysis and **b** optimized molecular structures of Ph-4PACz and Ph-Nd dimers. **c** Aromatic C-H bending vibration peaks in partial enlarged FTIR spectra of Ph-4PACz and Ph-Nd powders. **d** XPS spectra of C 1*s* for Ph-4PACz, Nd and Ph-Nd films on ITO. **e** Chemical shift variation of carboxyl proton in Nd, Ph-4PACz and Ph-Nd solutions. **f** XPS spectra of P 2*p* for Ph-4PACz and Ph-Nd films on ITO. **g** Phosphate characteristic peaks in partial enlarged FTIR spectra of Ph-4PACz and Ph-Nd powders.

measurements. All samples exhibit similar redox behavior without discrete stepwise oxidation features (Supplementary Fig. 3), indicating the formation of homogeneous SAMs with good molecular-level miscibility[30,31]. The highest occupied molecular orbital (HOMO) levels determined from CV were −5.25 and −5.17 eV for Ph-4PACz and Nd samples, respectively. The Ph-4PACz/Nd mixtures exhibit an evident nonlinear relationship between HOMO levels and blending ratios (−5.27 eV (1:0.5), −5.29 eV (1:1), and −5.26 eV (1:2))(Supplementary Fig. 4), which is further evidenced by ultraviolet photoelectron spectroscopy (UPS) results: −5.61 eV (1:0.5), −5.65 eV (1:1), and −5.62 eV (1:2) (Supplementary Figs. 5, 6, Supplementary Table 2). This nonlinear behavior could be mainly attributed to the frontier-orbital rearrangement between carbazole and indole moieties in the homogeneous SAMs[32–34] (Supplementary Figs. 6, 7). Moreover, the HOMO levels of the co-assembly SAMs align well with the valence band (−5.67 eV) of the perovskite absorber, favorable for boosting charge extraction and

photovoltaic performance. Based on these advantages (Fig. 1d), PSCs based on the 1:1 Ph-Nd yielded a maximum PCE exceeding 26%, whereas pure Nd-based PSCs suffered from a poor PCE below 20% (Supplementary Table 3). Therefore, the Ph-4PACz and 1:1 Ph-Nd were selected for further comprehensive investigation.

## Interactions within the co-adsorbed SAMs
To gain mechanistic insights into the homogeneous SAM formation, first-principles calculations were conducted. Independent gradient model based on Hirshfeld (IGMH) analysis reveals stronger intermolecular interactions existing in the Ph-Nd dimer than those in the Ph-4PACz dimer, as indicated by the increased density of blue scatter points[35] (Fig. 2a and Supplementary Fig. 8). The Ph-Nd dimer exhibits a shorter π-π stacking distance of 3.36 Å than 4.25 Å for the Ph-4PACz counterpart, whereas the Nd dimer adopted a disordered geometry configuration without π-π stacking, further affirming stronger π-π

interactions within Ph-Nd (Fig. 2b and Supplementary Fig. 9). Tetramer simulations provide additional evidence of improved molecular ordering, where the Ph-Nd adopted an alternating Ph-4PACz/Nd binding configuration in contrast to the disordered packing of the pure Ph-4PACz (Supplementary Fig. 10). Calculations based on SAM clusters attest a higher level of compactness and homogeneity in Ph-Nd compared with the Ph-4PACz (Supplementary Fig. 11). Furthermore, the Ph-Nd displays more negative binding energies at all aggregation levels—dimer (−0.83 eV), tetramer (−0.98 eV), and cluster (−2.82 eV), compared to the Ph-4PACz-only counterpart (−0.36, −0.57, and −1.77 eV) (Supplementary Fig. 12), signifying a strong thermodynamic driving force for the generation of co-adsorbed Ph-Nd. These findings suggest that Nd preferentially associates with Ph-4PACz, effectively inhibiting the Ph-4PACz self-aggregation while facilitating the formation of a highly ordered and uniformly distributed molecular arrangement in Ph-Nd (Supplementary Fig. 13).

To probe the solid-state structure, we performed X-ray diffraction (XRD) and Grazing Incidence Wide Angle X-ray Scattering (GIWAXS) measurements. XRD results confirm the amorphous nature of pure Ph-4PACz and Nd films, whereas the Ph-Nd film exhibits a sharp diffraction peak at 19.2° corresponding to the formation of a crystalline phase with a preferential orientation (Supplementary Fig. 14). A broad and isotropic halo observed in GIWAXS results further confirms an amorphous state of the pure Ph-4PACz film (Supplementary Fig. 15a). By contrast, the Ph-Nd film displays sharp diffraction arcs in the out-of-plane direction (Supplementary Fig. 15b), indicating the formation of a crystalline phase—highly ordered molecular arrangement. Moreover, the presence of a first-order peak at q = 0.69 Å$^{-1}$ and its clear second-order peak at 1.38 Å$^{-1}$ (i.e. the resulting interplanar spacing of 9.1 Å close to the thickness of the Ph-Nd dimer) indicate a long-range lamellar periodicity in the Ph-Nd film[36,37], which in turn evidenced the repetitive stacking of Ph-4PACz/Nd heterodimers (Supplementary Fig. 15c, d, Supplementary Table 4). Moreover, similar lamellar features with enhanced diffraction intensity were also observed at a higher precursor concentration of 4 mg/mL (Supplementary Fig. 16), confirming that the observed ordering is an intrinsic feature of the co-assembled phase across different film thicknesses. We further performed XRD measurements on recrystallized Ph-4PACz, Nd, and their 1:1 mixture (Ph-Nd) (Supplementary Fig. 17). Ph-4PACz and Nd showcase a dominant diffraction peak at 20.6° and 12.9°, respectively, while Ph-Nd exhibits a main diffraction peak at 19.2°. This diffraction shift indicates the formation of a new supramolecular phase rather than selective crystallization of either component, supporting molecular-level co-assembly and structural homogeneity in Ph-Nd.

To experimentally unravel the interaction mechanism driving the alternating co-adsorption of Ph-Nd, we carried out a series of characterization measurements including liquid-state nuclear magnetic resonance (NMR), Fourier transform infrared (FTIR), and X-ray photoelectron spectroscopy (XPS). In the ¹H NMR spectra (Supplementary Figs. 18–20), the proton signals on the carbazole moiety of Ph-4PACz and the indole unit of Nd exhibit distinct chemical shifts upon blending Ph-4PACz with Nd, while those of the alkyl chains remained unchanged (Δ <0.001 ppm) (Supplementary Figs. 21, 22). These chemical shift variations, primarily observed in the low-field region, indicate π-π interactions between the conjugated aromatic moieties of both Ph-4PACz and Nd[38]. These π-π interactions are further corroborated by FTIR results[39]. Specifically, compared with pure Ph-4PACz and Nd, the aromatic C-H bending peaks of the carbazole and indole moieties in Ph-Nd show a red shift and a blue shift of 2 cm$^{-1}$ (Fig. 2c and Supplementary Fig. 23), respectively. XPS analysis unveiled the π-π satellite peak at 291.38 eV in the C 1s spectrum of the Ph-4PACz film sharpened and intensified upon Nd incorporation, while this feature was absent in the Nd-only film (Fig. 2d), attesting enhanced π-π interactions in Ph-Nd[40,41]. In addition to the π-π stacking, spectroscopic analysis affirmed the existence of hydrogen bonding in Ph-Nd. Incorporating Nd into the Ph-4PACz solution resulted in a 0.04 ppm up-field shift and peak narrowing in the ³¹P NMR spectrum (Supplementary Fig. 24), and concurrently, the carboxyl proton peak in the ¹H NMR spectrum of the Nd solution disappeared upon Ph-4PACz addition (Fig. 2e), strongly evidencing the formation of hydrogen bonding between carboxyl and phosphate groups in Ph-Nd. The P 2$p_{1/2}$ and P 2$p_{3/2}$ binding energies of the Ph-Nd film relative to the Ph-4PACz film were reduced by 0.28 and 0.33 eV, respectively (Fig. 2f), and red/blue shifts of 2 cm$^{-1}$ in the phosphate-related stretching modes (P=O, P-O, P-O-H) for the Ph-4PACz powders were observed after Nd addition (Fig. 2g), further supporting the generation of hydrogen bonds in Ph-Nd[42,43]. Combining these experimental and computational results, we propose an alternating co-adsorption mechanism for Ph-Nd: Nd forms stable binding with Ph-4PACz through enhanced π-π interaction and hydrogen bonding, which not only suppresses Ph-4PACz self-aggregation but also enables the formation of highly uniform, alternating Ph-4PACz/Nd molecular arrangement in Ph-Nd.

## Homogeneity of SAM and perovskite films

To quantitatively compare the solution-state aggregation behavior for Ph-4PACz and Ph-Nd in methanol, dynamic light scattering (DLS) measurements were conducted (Fig. 3a, b). The freshly prepared Ph-Nd mixed solution shows a much smaller average particle size of 49 nm than that (118 nm) of the Ph-4PACz solution, representing the effective inhibition of Ph-4PACz self-aggregation after Nd addition. After 72 h of storage, the average colloidal size of the Ph-Nd solution grew moderately to 79 nm, whereas this colloidal size aggregated severely to 336 nm in the Ph-4PACz solution, further confirming the strong role of Nd against Ph-4PACz self-agglomeration. On this basis, the uniformity of Ph-4PACz and Ph-Nd films deposited on ITO substrates was investigated using scanning electron microscopy (SEM) equipped with energy-dispersive spectroscopy. The characteristic P element mapping demonstrates the existence of some obvious aggregation and vacancy regions in the Ph-4PACz film, whereas the P element is uniformly distributed in the Ph-Nd hybrid film (Supplementary Fig. 25). This enhanced SAM's homogeneity is further verified by atomic force microscopy (AFM) results (Supplementary Fig. 26), where the Ph-Nd film presents a smaller root-mean-square (RMS) roughness of 1.48 nm relative to 1.73 nm for the Ph-4PACz counterpart. Additionally, AFM-based infrared spectroscopy (AFM-IR) results directly evidenced that the Ph-4PACz film became much more uniform after Nd incorporation according to the characteristic signal of P-O stretching vibration (Fig. 3c, d).

To further quantitatively analyze the SAM coverage, the coverage factor based on the XPS peak area ratio of P 2p and In 3$d_{3/2}$ was determined to be $4.94 \times 10^{-2}$ and $6.89 \times 10^{-2}$ for the Ph-4PACz and Ph-Nd films, respectively, indicating a higher Ph-4PACz adsorption density in the alternating co-adsorbed SAM (Supplementary Fig. 27). Such a higher adsorption density is further evidenced by CV analysis (Supplementary Fig. 28): the areal density of Ph-4PACz in the Ph-Nd film is $1.34 \times 10^{13}$ molecules cm$^{-2}$, much larger than the $1.26 \times 10^{13}$ molecules cm$^{-2}$ for the Ph-4PACz film[16,44]. To evaluate the carrier transport properties of different SAMs, the hole-only devices with an architecture of ITO/SAM/Cu were manufactured based on the space-charge-limited current (SCLC) model. The Ph-Nd film exhibits higher conductivity and hole mobility than those of the Ph-4PACz film based on the slope analysis (Supplementary Fig. 29a, b). Besides, SEM images indicate a more distinct view of the ITO surface observed for the Ph-Nd film in contrast with the Ph-4PACz sample (Supplementary Fig. 29c, d), further supporting higher surface charge transfer induced by Nd treatment[45]. The optimized carrier transfer for the alternating co-adsorbed SAM is attributed to enhanced phase homogeneity, which promotes efficient hole extraction at the buried heterointerface between perovskite and SAM.

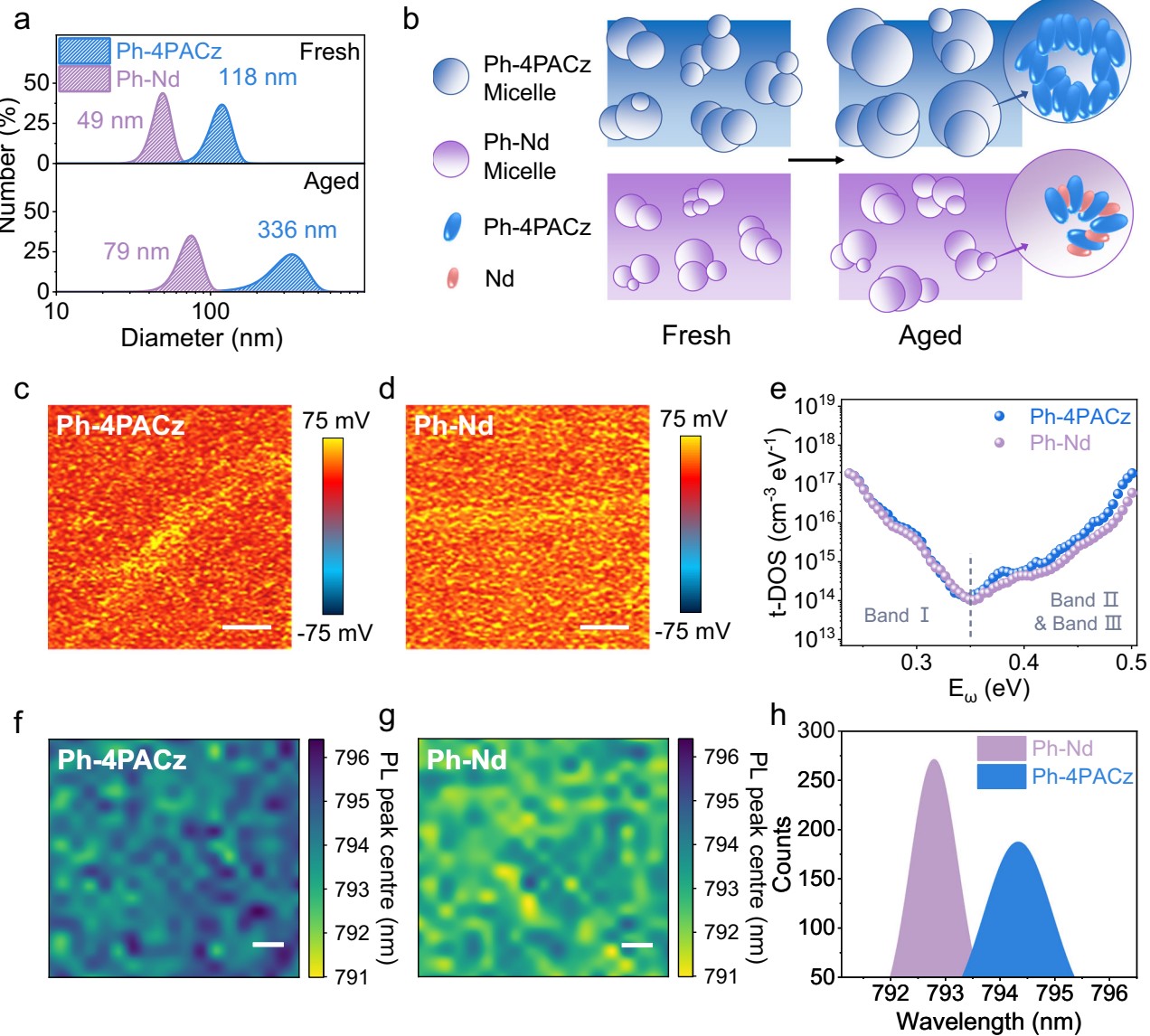

**Fig. 3 | Homogeneity of SAM and perovskite films. a** Particle size distributions of Ph-4PACz and Ph-Nd dissolved in methanol before and after 72 h storage. **b** Schematic illustration of molecular micelles before and after aging. AFM-IR images of **c** Ph-4PACz and **d** Ph-Nd films deposited on ITO (scale bar: 1 μm), depending on the feature peak of P-O stretching vibration. **e** t-DOS plots of full devices based on Ph-4PACz and Ph-Nd. PL peak location mapping of encapsulated perovskite films grown on **f** Ph-4PACz and **g** Ph-Nd substrates (scale bar: 1 μm). **h** Statistical distributions of PL peak locations.

To elucidate the impact of Nd addition on the interactions between SAM and perovskite, several characterization techniques containing FTIR, NMR (Supplementary Figs. 30–32) and XPS were employed. FTIR results illustrate mingling Nd with $PbI_2$ led to no obvious shift in the C=O stretching vibration, implying almost no interaction existing between Nd and $PbI_2$ (Supplementary Fig. 33). By contrast, replacing $PbI_2$ with FAI resulted in an 8 cm$^{-1}$ red shift and a 3 cm$^{-1}$ blue shift in the C=O and N-H stretching vibrations (Supplementary Fig. 34), respectively, corresponding to the formation of C=O···H-N hydrogen bonding between Nd and FAI. Moreover, the carboxyl proton peak of the Nd solution shows an up-field shift of 0.002 ppm upon FAI addition, whereas no chemical shift was observed in the mixed Nd/$PbI_2$ solution (Supplementary Fig. 35). Simultaneously, the amino proton peak at 8.834 ppm of the FAI solution split into multiple peaks upon Nd inclusion. These NMR results further validate the generation of C=O···H-N hydrogen bonds between Nd and FAI. This hydrogen bonding is further proved by XPS results. Inserting

a Ph-4PACz layer between perovskite and ITO produced a red shift of Pb 4$f$ peaks by 0.03 eV, however, the further incorporation of Nd into SAM resulted in no detectable chemical shift (Supplementary Fig. 36). Likewise, the perovskite film grown on Ph-Nd showcases no detectable shift in the I 3$d$ peaks relative to that deposited on Ph-4PACz. Contrastingly, a blue shift of 0.06 eV in the N 1$s$ peak of the perovskite film on Ph-Nd relative to that on Ph-4PACz was observed (Supplementary Fig. 37).

SEM characterization was implemented to assess the effect of different SAMs on perovskite homogeneity. The perovskite film deposited on Ph-Nd shows a more uniform surface morphology compared with that deposited on Ph-4PACz, where the distribution range of the grain size extracted from the top surface immediately narrowed down from 715 to 476 nm upon Nd addition, which was also observed in the buried surface (Supplementary Figs. 38, 39). Moreover, XRD results unveiled the peak intensity proportion of the dominant {100} crystal plane family of the perovskite film deposited on

Ph-4PACz increased from 54% to 66% when inserting Nd, signifying enhanced uniformity of the perovskite orientation grown on Ph-Nd (Supplementary Fig. 40a–c and Table 5). Benefitting from improved perovskite homogeneity, the perovskite film coated on Ph-Nd showcases a smaller residual stress of $4.2 \times 10^{-2}$ relative to that grown on Ph-4PACz ($7.8 \times 10^{-2}$) (Supplementary Fig. 40d). The improved perovskite homogeneity is in good accordance with thermal admittance spectroscopy results (Fig. 3e). The incorporation of Nd into Ph-4PACz led to a decline in the trap density of states (t-DOS) for full devices, particularly for deep-level defects existing at grain surfaces which inevitably mitigates non-radiative charge recombination through defects[46,47]. Subsequently, PL mapping measurements were performed to further evaluate the spatial distribution of perovskite materials grown on different SAMs. The perovskite film deposited on Ph-Nd exhibits a narrower distribution range of PL peak locations in parallel with higher and more uniform PL intensities in contrast with that coated on Ph-4PACz (Fig. 3f–h and Supplementary Fig. 41), evidencing better optoelectronic homogeneity of the perovskite film grown on Ph-Nd[48]. The red shift in the PL peak by ~1.5 nm observed for perovskite on Ph-4PACz likely originates from the generation of separated or defective phases[18,49].

To delve into the impact of perovskite homogeneity on charge transfer and recombination dynamics, the bi-exponential fitting of time-resolved PL (TRPL) spectra revealed that the perovskite film grown on Ph-Nd exhibits a shorter charge transfer lifetime ($\tau_1$) of 1.48 μs and a longer charge recombination lifetime ($\tau_2$) of 17.76 μs, compared to that on Ph-4PACz (1.69 and 12.68 μs) (Supplementary Fig. 42 and Supplementary Table 6), indicating the Nd inclusion effectively accelerates initial charge extraction and restrains non-radiative recombination[50]. Based on TRPL results, the surface recombination velocity (SRV) was calculated to be 0.15 and 0.07 m s$^{-1}$ for perovskite films coated on Ph-4PACz and Ph-Nd, respectively, suggesting the interfacial non-radiative recombination can be efficiently inhibited by Nd addition[51]. The perovskite films on Ph-4PACz and Ph-Nd yielded a PL quantum yield (PLQY) value of 4.52% and 5.89%, respectively (Supplementary Fig. 43), corresponding to non-radiative recombination loss on open-circuit voltage ($\Delta V_n$) of 80.07 and 73.23 mV, representing reinforced radiative recombination induced by Nd incorporation. Such stronger radiative recombination is further validated by quasi-Fermi level splitting (QFLS) results (Supplementary Fig. 44), delivering a QFLS value of 1.206 and 1.213 eV for perovskite films on Ph-4PACz and Ph-Nd, respectively. To further quantify the trap states of perovskite films deposited on different SAMs, SCLC analysis indicated the resultant defect density diminished from $6.32 \times 10^{15}$ to $5.28 \times 10^{15}$ cm$^{-3}$ after Nd addition (Supplementary Fig. 45).

## Device performance

To evaluate the impact of different SAMs on device performance, inverted PSCs with a configuration of ITO/SAM/perovskite/C$_{60}$/bathocuproine (BCP)/Cu were fabricated (Fig. 4a). As depicted in Fig. 4b and Supplementary Table 7, the Ph-Nd-based device achieved a maximum PCE of 26.95% under reverse scan, accompanying with an open-circuit voltage ($V_{OC}$) of 1.185 V, a short-circuit current density ($J_{SC}$) of 26.16 mA cm$^{-2}$, and a fill factor (FF) of 86.93%, which outperformed the Ph-4PACz counterpart (PCE = 25.12%, $V_{OC}$ = 1.174 V, $J_{SC}$ = 25.74 mA cm$^{-2}$, and FF = 83.14%). The statistical analysis of photovoltaic parameters further confirmed better performance of the Ph-Nd device than the Ph-4PACz counterpart (Supplementary Fig. 46 and Supplementary Table 7). Both devices exhibited negligible hysteresis of J-V characteristics (Supplementary Fig. 47 and Supplementary Table 8). External quantum efficiency (EQE) spectra were recorded to verify the $J_{SC}$ (Supplementary Fig. 48), which yielded an integrated $J_{SC}$ of 25.60 mA cm$^{-2}$ for the Ph-Nd device, congruent with that extracted from J-V measurements. The stabilized power output was measured at

the maximum power point tracking (MPPT) over 600 s (Supplementary Fig. 49), delivering a stabilized PCE of 26.52% and 24.43% for the Ph-4PACz and Ph-Nd devices, respectively. Notably, our champion device achieved a certified PCE of 26.57% along with an MPPT efficiency of 26.23% (Supplementary Fig. 50). This high PCE can be mainly ascribed to enhanced phase homogeneity with accelerated charge extraction and inhibited non-radiative recombination. To further demonstrate the benefit of phase homogenization induced by our alternating co-adsorption strategy, the large-area devices with an active area of 1 cm$^2$ were manufactured (Fig. 4c, Supplementary Fig. 51 and Supplementary Table 9). The Ph-Nd device realized a peak PCE of 25.61%, highlighting high quality of the co-assembled SAM, much higher than 24.03% of the Ph-4PACz device. To showcase the universality of our alternating co-adsorption methodology, the Nd was further mixed with other carbazole-based SAMs including 2PACz and (2-(3,6-dimethoxy-9H-carbazole-9-yl)ethyl)phosphonic acid (MeO-2PACz), which resulted in prominent improvements in photovoltaic performance compared to untreated devices (Fig. 4d and Supplementary Fig. 52). In addition, the alternating co-adsorbed SAM strategy was also applicable to wide band-gap PSCs (1.68 and 1.78 eV), which effectively elevated device performance (Fig. 4e and Supplementary Fig. 53).

To unravel the origin of performance enhancements induced by Nd modification, the charge recombination dynamics of devices were analyzed and compared. The slope of $V_{OC}$ versus the natural logarithm of light intensity produced an ideality factor of 1.42 for the Ph-4PACz device, which declined to 1.25 for the Ph-Nd device (Supplementary Fig. 54), evidencing more effective restraint of trap-assisted non-radiative recombination in the Ph-Nd device[52]. The Mott-Schottky (MS) analysis revealed a higher built-in potential ($V_{bi}$) of 1.02 V for the Ph-Nd device than 0.77 V for the Ph-4PACz device (Supplementary Fig. 55), indicative of a stronger internal field existing in the Ph-Nd device that expedites charge separation and accordingly raises $V_{OC}$[49]. Transient photocurrent (TPC) and transient photovoltage (TPV) measurements (Supplementary Fig. 56) further confirmed a faster charge extraction lifetime of 474.20 ns and a longer charge recombination lifetime of 1.09 μs for the Ph-Nd device, relative to 705.57 ns and 0.71 μs for the Ph-4PACz device. Such improved charge transfer and recombination coincide well with dark J-V measurements (Supplementary Fig. 57), where the Ph-Nd device has a smaller leakage current than the Ph-4PACz device. After the fitting of Nyquist plots, the Ph-4PACz and Ph-Nd devices delivered a charge transfer resistance ($R_{ct}$) of 383.5 and 115.4 Ω, respectively, together with a charge recombination of resistance ($R_{rec}$) of 9513 and 16877 Ω (Supplementary Fig. 58 and Table 10), suggesting the Nd inclusion speeds up the charge extraction and slows down the charge recombination. Eventually, the FF loss analysis illustrates that both non-radiative and transport recombination losses were reduced in the Ph-Nd device compared to the Ph-4PACz device (Supplementary Fig. 59), therefore leading to an enhanced FF for the Ph-Nd device.

Device stability was further assessed to examine the impact of different SAMs on the long-term operational performance under different standard protocols (Fig. 4f, g). The Ph-Nd-based device retained 93.36% of its initial PCE after 1500 h MPPT under continuous one-sun light-emitting diode (LED) illumination in a N$_2$ atmosphere, whereas the Ph-4PACz device shows only 78.01% retention. In parallel, thermal stability measurements discovered that the Ph-Nd device maintained 91.10% of its original PCE after 1200 h heating at 85 °C in nitrogen, surpassing 72.68% of the Ph-4PACz device. This enhanced stability is highly associated with the uniform and dense alternating co-adsorbed SAM layer, which exhibits enhanced intrinsic thermal and electronic stability compared to the pure Ph-4PACz film (Supplementary Figs. 60, 61). The broad applicability and benefits of our alternating co-adsorption were summarized in Fig. 4h.

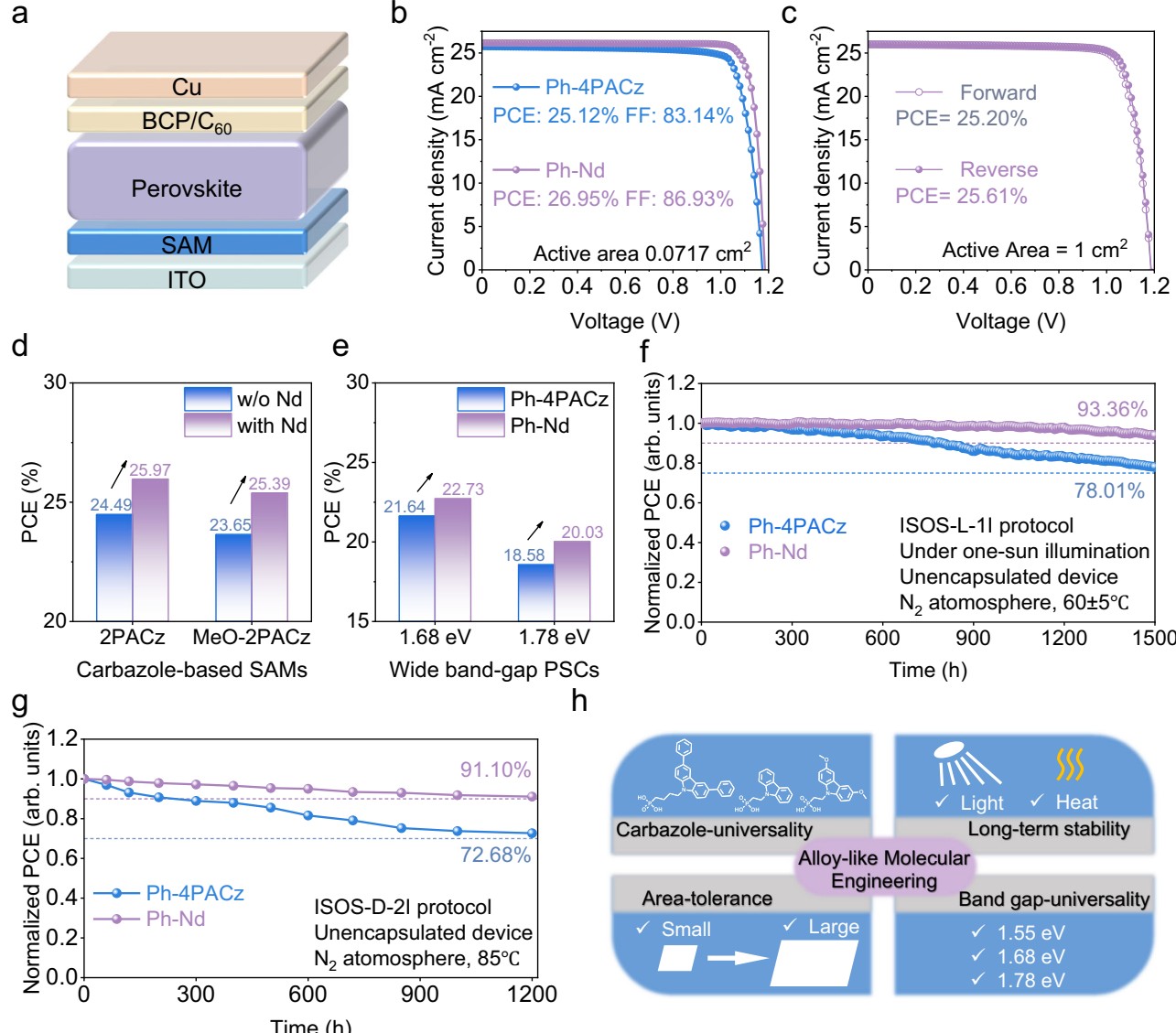

**Fig. 4 | Device performance. a** Schematic diagram of device architecture. **b** *J-V* curves of 0.0717 cm² champion PSCs based on Ph-4PACz and Ph-Nd under reverse scan. **c** *J-V* curves of 1 cm² Ph-Nd PSCs under forward and reverse scan. **d** PCE comparison of PSCs based on 2PACz and MeO-2PACz without and with Nd. **e** PCE comparison of wide band-gap PSCs based on Ph-4PACz and Ph-Nd. **f** Operational stability of unencapsulated devices at the MPPT under one-sun LED illumination in N₂ (initial: 24.73% for Ph-4PACz and 26.69% for Ph-Nd). **g** Thermal stability of unencapsulated devices under constant heating at 85 °C in N₂ (initial: 24.89% for Ph-4PACz and 26.78% for Ph-Nd). **h** Summary of advantages of the alternating co-adsorption molecular engineering.

## Discussion

In summary, we demonstrate an indole-carbazole alternating co-adsorption strategy to fabricate uniform SAMs by introducing Nd into Ph-4PACz. It is found that Nd binds preferentially with Ph-4PACz through enhanced π-π interactions and hydrogen bonding. This alternating architecture not only suppresses Ph-4PACz self-aggregation and enables the formation of highly uniform, alternating Ph-4PACz/Nd molecular arrangement in Ph-Nd, but also features nonlinear energy levels to realize well-suited alignment at the SAM and perovskite interface. This alternating co-adsorption strategy also prominently improves the homogeneity of perovskite films, resulting in faster charge transport and reduced non-radiative recombination losses. As a result, the small-area (0.0717 cm²) devices yield a maximum PCE of 26.95% (certified 26.57%), in parallel with a PCE of 25.61% over 1 cm². These devices retain 93.36% of their initial PCEs after 1500 h MPPT under continuous illumination and 91.10% after 1200 h

heating at 85 °C. The broad applicability of this alternating co-adsorption strategy across carbazole-based SAM systems and wide band-gap PSCs establishes it as a fundamental framework for molecular interface engineering in high-performance PSCs.

## Methods

### Materials

All chemical materials purchased from commercial suppliers were directly employed without further purification. Nd, Ph-4PACz, MeO-2PACz, guanidine thiocyanate (GuaSCN, >99.5%) and 1,3-diaminopropane dihydroiodide (PDADI₂) were bought from Xi'an Yuri Solar Co., Ltd (China). Lead (II) iodide (PbI₂, 99.99%) was procured from TCI (Japan). Formamidine iodide (FAI, 99.5%), methylammonium iodide (MAI, 99.5%), methylammonium chloride (MACl, 99.5%), and cesium iodide (CsI, 99.99%), C₆₀ and BCP were observed from Advanced Election Technology Co., Ltd. (China). Dimethyl sulfoxide (DMSO, 99.7%), N,N-dimethyl formamide (DMF, 99.8%), chlorobenzene (CB,

99.9%), Methanol (MT, 99.9%) and isopropyl alcohol (IPA, 99.9%) were provided from Aladdin (China).

## Preparation of perovskite precursor solutions

The 1.5 M $FA_{0.85}MA_{0.1}Cs_{0.05}PbI_3$ perovskite precursor was obtained by completely dissolving stoichiometric $PbI_2$, FAI, MAI, and CsI in a mixed solvent (DMF/DMSO = 4:1 in volume). For further modifying the precursor solution, 10 mg mL$^{-1}$ MACl and 8 mg mL$^{-1}$ GuaSCN were incorporated into the precursor. The solution was magnetically stirring at room temperature until completely dissolved, and then it was filtered through a 0.22 μm polytetrafluoroethylene (PTFE) membrane. Wide band-gap perovskite compositions adopted in this work were $FA_{0.8}Cs_{0.05}MA_{0.15}Pb(I_{0.755}Br_{0.255})_3$ (1.68 eV) and $FA_{0.8}Cs_{0.2}PbI_{1.8}Br_{1.2}$ (1.78 eV), and the corresponding precursor (1.4 M) was prepared as described above.

## Fabrication of perovskite solar cells

ITO substrates ($2 \times 2$ cm$^2$) were performed with the ultrasonic cleaning in detergent, DI water, acetone, and isopropanol (30 min each) sequentially, followed by a 30 min ultraviolet-ozone treatment before transferring to a nitrogen-filled glovebox. The methanol solutions (1 mL) containing 0.001 mmol Ph-4PACz (with and without 0.001 mmol Nd) was prepared as SAM solution. 80 μL SAM solution was spin-coated on cleaned ITO substrates at 4500 rpm for 30 s, and the wet substrates were then annealed at 100 °C for 15 min. The 90 μL perovskite precursor was then spin-coated onto SAM substrates via a two-step program (1000 rpm for 10 s, 5000 rpm for 30 s), where the 150 μL chlorobenzene was dripped onto substrates at 10 s before the spin-coating completion. Subsequently, the wet perovskite film deposited on substrates was heated at 100 °C for 15 min, followed by spin-coating the $PDADI_2$ solution (0.3 mg/mL in IPA) at 5000 rpm for 30 s and a successive heating at 100 °C for 5 min. The $C_{60}$ (25 nm) and BCP (11 nm) layers were sequentially deposited via thermal evaporation, followed by depositing Cu electrode through a metal mask (100 nm, $6 \times 10^{-4}$ mba).

## Characterization of thin films

UV–Vis was measured by SHIMADZU CORP UV-2600i. CV measurements were performed on a Chenhua CHI660E electrochemical workstation equipped with a three-electrode system. UPS and XPS measurements were performed using a vacuum surface analyzer comprising an ULVAC-PHI 5000 VersaProbe III spectrometer, with the He I (21.22 eV) excitation source for UPS and monochromatic Al Kα radiation (1486.6 eV) for XPS. FTIR spectra was recorded by VERTEX 70 and Tensor II (Bruker). NMR spectra were measured on Bruker AVANCE III 400 M. SEM measurements were performed at 5.0 kV (JEOL JSM-7610F Plus), Nippon Electronics Corporation. AFM was captured by Oxford Instruments Asylum Research. AFM-based infrared spectroscopy was performed by Bruker Icon. XRD measurements were carried out using a D8 ADVANCE XRD spectrometer (Bruker) with a Cu Kα line of λ = 1.5410 Å. GIWAXS measurements were carried out on the Synchrotron and Printable Electronics lab at Shenzhen Polytechnic with SaxsFocus using a Cu X-ray source (8.05 keV, 1.54 Å) and a Pilatus3R 300 K detector. PL mapping images were obtained via LabRAM Odyssey (HORIBA FRANCE SAS), and the excitation wavelength was 532 nm. TRPL was analyzed by the FLS 1000 fluorescence spectrometer equipped with a 450 nm excitation.

## Characterization of devices

A Keithley 2400 Source Measure Unit (SMU) was employed to characterize J-V curves and stabilized power output (SPO) in a N$_2$-filled glovebox under AM 1.5 G illumination (Newport-Oriel Sol3A) and J-V characteristics under dark, where the aperture of metal masks was used to accurately define the active area of solar cells. EQE spectra were measured using an Enlitech QE-R quantum efficiency system. MS curves and EIS were recorded on a Chenhua CHI660E electrochemical workstation. For long-term operational stability, unencapsulated devices were subjected to continuous illumination under a 1-sun equivalent light-emitting diode (LED) lamp in a N$_2$ environment (91PVKSolar Co. Ltd), and the evolution of device performance was monitored by the MPPT system (Shenzhen Lancheng Technology Co. Ltd). Thermal stability tests involved exposing unencapsulated device sets (6 samples per group) to 85 °C in N$_2$ and evaluating them at 48-h intervals.

## Computational details

Theoretical calculations were carried out using Gaussian 09. The Ph-4PACz and Nd were optimized at the B3LYP/6-311 G(d,p) level, followed by frequency calculations. For the treatment of periodic systems, the Vienna Ab Initio Simulation Package (VASP) was employed. The calculations in VASP adopted the PBE functional, with the plane-wave cutoff energy set to ENCUT = 450 eV, the electronic self-consistent field (SCF) convergence threshold set to $1 \times 10^{-5}$, and the maximum force convergence criterion set to 0.02 eV Å$^{-1}$.

## Reporting summary

Further information on research design is available in the Nature Portfolio Reporting Summary linked to this article.

## Data availability

All data generated or analyzed during this study are included in the published article and its Supplementary Information. Additional data are available from the corresponding author on request. Source data are provided with this paper.

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

## Acknowledgements

This work was financially supported by the National Natural Science Foundation of China (62004089 and 62374053), the Basic Research Project of Key Scientific Research Programs in Henan Higher Education Institutions (26ZX006), the Guangdong Basic and Applied Basic Research Foundation (2024A1515140075), and the Special Zone Support Program for Outstanding Talents of Henan University.

## Author contributions

H.S., Y.J., F.L. and K.Z. contributed equally to this work. S.C. conceived and designed the research. S.C., M.K.N., X.W. and G.Q. supervised the project. H.S. performed spectroscopy measurements, DFT calculation and analyzed the results with the supervision from S.C. and N.S. Y.J. prepared the small and large-area devices with the guidance from S.C. and G.Q. K.Z. contributed to conduct morphology analysis. J.Y. and Y.L. conducted PLQY and QFLS measurement. Z.L. and Y.S. helped with the stability testing. H.L. and F.G. contributed to the characterization of wide-gap PSCs. H.S. and F.L. wrote the first draft. X.W., Z-X.X. and M.K.N. contributed to provide advice. S.C., G.Q. and Y.D. finalized the manuscript. All authors contributed to the formal analysis and discussion of the data.

## Competing interests

The authors declare no competing interests.
