## [Transparent Peer Review file · Nature Communications]

Homogenizing Interfacial Assembly via Indole-Mediated Binary Monolayers for Perovskite Solar Cells

Corresponding Author: Professor MOHAMMAD KHAJA Nazeeruddin

Version 0:

Reviewer comments:

Reviewer #1

(Remarks to the Author)

In this manuscript, the authors reported an indole-carbazole alloy-like strategy to fabricate uniform SAMs by introducing Nd into Ph-4PACz. They proposed alloy-like architecture not only suppressed Ph-4PACz self-aggregation and enabled the formation of highly uniform, alternating Ph-4PACz/Nd molecular arrangement in Ph-Nd, but also featured nonlinear energy levels to realize well-suited alignment at the SAM/perovskite interface. Finally, the small-area devices yielded a maximum PCE of 26.95% (certified 26.57%). These devices retained 93.36% of their initial PCEs after 1500 h MPPT under continuous illumination and 91.10% after 1200 h heating at 85 °C. However, there are still some unclear points in this study. The manuscript needs a major revision with following concerns.

1. Why called Alloy-like Self-selected Layers? Was there any direct experimental evidence to prove the alternating arrangement of SAMs in the fig.1 D?
2. How about the photostability of Nd?
3. Can you provide detailed steps for CV testing of Ph-4PACz films without and with Nd?
4. The UPS results of ITO/SAM are generally considered to be related to the dipole moment of SAM, how did frontier-orbital rearrangement between carbazole and indole moieties in the alloy-like SAMs specifically affect the energy levels measured based on UPS, what was the ITO energy level based on UPS measurement.
5. The freshly prepared Ph-Nd mixed solution shows an average particle size of 49 nm, this seemed to be more than just the main dimers, trimers or pentamers shown in Fig.3 b.
6. How enhanced intermolecular interactions affect assembly behavior compared to using mixed solvents to adjust dispersibility.
7. Why was Ph-4PACz to Nd 1 ratio 1 the best, Nd seemed to be more than Ph-4PACz in Fig.3 b.

Reviewer #2

(Remarks to the Author)

This manuscript describes a Ph-4PACz/Nd mixed molecular layer formed by introducing N-indoleacetic acid into the well-known Ph-4PACz system, used here as a self-assembled monolayer in perovskite solar cells. The authors suggest that the two molecules interact through hydrogen bonding and π - π stacking, helping to reduce SAM self-aggregation. However, the central claim of an “alloy-like” small-molecule arrangement—an alternating molecular assembly illustrated in Fig. 1 and Supplementary Fig. 13—remains unsupported, as no structural evidence (such as XRD or GIWAXS) is provided. The interactions they point out may indeed exist, but the evidence presented is indirect and does not demonstrate the proposed molecular ordering.

Furthermore, many SAM systems based on similar π - π and hydrogen-bond interactions have already been reported, several of which also suppress aggregation effectively. (e.g., Nature Chemistry, 2025, <https://doi.org/10.1038/s41557-025-01732-z>, Nature Photonics 2025, 19, 1070–1077, Nat. Synth (2025). <https://doi.org/10.1038/s44160-025-00896-3>, Nat. Photon. 2024, 18, 1269–1275). As a result, the novelty here is rather limited, and the proposed mechanism does not show a clear advantage over existing systems. For these reasons, I am unable to recommend this work for publication in Nature Communications.

Reviewer #3

(Remarks to the Author)

This manuscript reports an alloy-like self-assembled monolayer (SAM) strategy achieving a certified efficiency exceeding stabilized 26% at Fujian Metrology Institute, which represents a highly robust result. The overall testing and characterization are relatively comprehensive, and the work is substantially complete. Therefore, I recommend "Accept after Minor Revision." However, several critical issues require clarification and improvement:

1. The somewhat vague interpretation of certain characterization results and insufficient logical connections. For instance, the link between UV-Vis/PL results and SAM aggregation (ordered vs. disordered aggregation) needs stronger evidence; the current data does not directly demonstrate this relationship. Additionally, the claim on Page 7, Line 179 ("These π - π interactions are further corroborated by FTIR results") requires explicit elaboration of the specific FTIR evidence. Furthermore, the rationale for using ethylene glycol (EG) and water in contact angle experiments to reflect miscibility is unclear, especially since the SAM is reportedly soluble in methanol – this methodological choice needs justification.
2. Verification of molecular ordering via techniques like XRD or GIWAXS is strongly recommended to substantiate this claim. Moreover, the stability of this SAM layer itself might be the primary factor contributing to the enhanced overall device stability. Can the authors discuss this potential causality. What are the initial efficiency values of the devices in the stability test results (Figs. 4f and 4g)?
3. The noise level in the tDOS data (Fig. 3e) is excessively high, potentially compromising its reliability. Re-testing this data is advised to ensure accuracy.
4. The methodology for mobility testing (Fig. S25) needs clarification. The specific model used and confirmation of Ohmic contact formation in the devices must be detailed. I suggest that a qualitative description might be more appropriate than a quantitative claim.
5. The discussion regarding the impact of ordered vs. disordered SAM substrates is contentious within the field. The Introduction section is suggested to include this discussion, and some representative prior work should be cited to provide context for the current study such as Nat. Chem. 17, 564–570 (2025). <https://doi.org/10.1038/s41557-025-01732-z>; Nat. Photon. 18, 1269–1275 (2024). <https://doi.org/10.1038/s41566-024-01531-x>; Nat. Photon. (2025). <https://doi.org/10.1038/s41566-025-01778-y>.

Version 1:

Reviewer comments:

Reviewer #1

(Remarks to the Author)

The authors have provided additional experimental data and explanation on the reviewers' comments. Now, the manuscript can be accepted for publication.

Reviewer #2

(Remarks to the Author)

I am still not convinced that this system can be described as an "alloy." Based on the current evidence, this appears to be a mixed or co-adsorbed SAM rather than an alloy-like phase. I therefore suggest replacing "alloy-like" with "alternating co-adsorbed SAM" or simply "mixed SAM," as the present terminology is conceptually misleading. If the authors wish to retain the term "alloy," they need to provide stronger and more direct evidence consistent with a solid-solution like assembly. At a minimum, this would require a composition-dependent XRD series over several Ph-4PACz:Nd ratios, showing a continuous evolution of the diffraction features (e.g., systematic peak shifts). In addition, independent evidence excluding phase separation is essential, for example via nanoscale chemical mapping (such as ToF-SIMS, spatially resolved XPS, or AFM-IR) demonstrating molecular-level homogeneity across the SAM.

Moreover, the GIWAXS section lacks critical experimental details. For organic SAMs, it is inherently difficult to obtain clear diffraction features because of their ultrathin nature. It is therefore unclear how such pronounced out-of-plane arcs were obtained here. Were these measurements performed on monolayers on ITO, or on much thicker spin-cast films? If elevated concentrations or thick films were required to enable GIWAXS detection, the observed lamellar stacking (as suggested by the q_z arcs) likely reflects an aggregated or bulk-like state, which may be fundamentally different from the interfacial SAM present in actual devices. In that case, the structural conclusions would not be transferable to the operational devices. The authors should describe how the GIWAXS samples were prepared and compare these conditions with those used for device fabrication, making them consistent. Otherwise, the structural picture may correspond to a different system. Given that these GIWAXS results are treated as a core mechanistic proof, they also deserve more space and a deeper discussion in the main text.

Finally, the emergence of new diffraction peaks raises another concern: if the initial concentrations are comparable to those used for the single-component films, do these peaks reflect templated crystallization of one specific molecule rather than a genuinely mixed phase? This possibility should be addressed.

Reviewer #3

(Remarks to the Author)

I have no further comments; I believe the current version is ready for publication.

Version 2:

Reviewer comments:

Reviewer #2

(Remarks to the Author)

The revised manuscript reflects improved scientific rigor and strengthened innovation. All major concerns have been satisfactorily addressed. I have no further comments and recommend acceptance as submitted.

REVIEWER COMMENTS

Reviewer #1:

In this manuscript, the authors reported an indole-carbazole alloy-like strategy to fabricate uniform SAMs by introducing Nd into Ph-4PACz. They proposed alloy-like architecture not only suppressed Ph-4PACz self-aggregation and enabled the formation of highly uniform, alternating Ph-4PACz/Nd molecular arrangement in Ph-Nd, but also featured nonlinear energy levels to realize well-suited alignment at the SAM/perovskite interface. Finally, the small-area devices yielded a maximum PCE of 26.95% (certified 26.57%). These devices retained 93.36% of their initial PCEs after 1500 h MPPT under continuous illumination and 91.10% after 1200 h heating at 85 °C. However, there are still some unclear points in this study. The manuscript need a major revision with following concerned issues.

Response:

Thank you for your positive feedback and constructive suggestions. We sincerely appreciate your recognition of the molecular-scale insights provided by our alloy-like SAM strategy in rationalizing device performance. In response to your comments, we have carefully revised the manuscript and addressed all points raised in detail. We believe these revisions further highlight the broader implications of our findings for rational molecular and interfacial material design, thereby strengthening the overall impact of the manuscript.

Comment 1: Why called Alloy-like Self-selected Layers? Was there any direct experimental evidence to prove the alternating arrangement of SAMs in the fig.1 D?

Response:

Thank you for your insightful and fundamental questions, which helped us to clarify the conceptual framework and provide additional supporting evidence.

On the rationale for “Alloy-like Self-Assembled Monolayers”:

The term “alloy-like” is used to describe the molecular-level miscibility and uniform co-assembly of two structurally analogous yet chemically distinct components—Ph-4PACz and Nd—into a homogeneous SAM. This is conceptually analogous to metallic alloys, in which distinct atoms form a single-phase solid solution. The alloy-like behavior in our system arises from strong directional interactions (π - π stacking and hydrogen bonding) between the carbazole and indole units, as revealed by first-

principles calculations and supported by spectroscopic signatures (NMR, FTIR, XPS), as well as nonlinear HOMO evolution from CV and UPS results. These interactions promote self-selection and energetic stabilization of a mixed packing state.

To better reflect the core of this molecular engineering strategy, we have revised the manuscript title to: “Alloy-like Molecular Engineering of Self-Assembled Monolayers for Efficient and Stable Inverted Perovskite Solar Cells”.

Direct evidence for the alternating arrangement:

To further substantiate the proposed alternating Ph-4PACz/Nd configuration, we conducted additional XRD and GIWAXS measurements. XRD analysis (Fig. R1) revealed that the pure Ph-4PACz and Nd films were amorphous, whereas the Ph-Nd film exhibits a sharp diffraction peak at 19.2° corresponding to the generation of a crystalline phase with a preferential orientation. A broad and isotropic halo observed in GIWAXS results further confirmed an amorphous state of the pure Ph-4PACz film (Fig. R2a). By contrast, the Ph-Nd film displays sharp and strong diffraction arcs in the out-of-plane direction (Fig. R2b), indicating the formation of a crystalline phase—highly ordered molecular arrangement. Moreover, the presence of a first-order peak at $q = 0.69 \text{ \AA}^{-1}$ and its clear second-order peak at 1.38 \AA^{-1} (i.e. the resulting interplanar spacing of 9.1 \AA close to the thickness of the Ph-Nd dimer) indicated a long-range lamellar periodicity in the Ph-Nd film, which in turn evidenced the repetitive stacking of Ph-4PACz/Nd heterodimers (Fig. R2c-d, Table R1). These results strongly confirmed the alternating molecular arrangement in Ph-Nd.

We have added the follow discussion on page 7 line 176 of the revised manuscript: “To probe the solid-state structure, we performed X-ray diffraction (XRD) and Grazing Incidence Wide Angle X-ray Scattering (GIWAXS) measurements. XRD results confirmed the amorphous nature of pure Ph-4PACz and Nd films, whereas the Ph-Nd film exhibits a sharp diffraction peak at 19.2° corresponding to the formation of a crystalline phase with a preferential orientation (Supplementary Fig. 14). A broad and isotropic halo observed in GIWAXS results further confirmed an amorphous state of the pure Ph-4PACz film (Supplementary Fig. 15a). By contrast, the Ph-Nd film displays sharp and strong diffraction arcs in the out-of-plane direction (Supplementary Fig. 15b), indicating the formation of a crystalline phase—highly ordered molecular arrangement. Moreover, the presence of a first-order peak at $q = 0.69 \text{ \AA}^{-1}$ and its clear second-order peak at 1.38 \AA^{-1} (i.e. the resulting interplanar spacing of 9.1 \AA close to

the thickness of the Ph-Nd dimer) indicated a long-range lamellar periodicity in the Ph-Nd film, which in turn evidenced the repetitive stacking of Ph-4PACz/Nd heterodimers (Supplementary Fig. 15c-d, Supplementary Table 4).” Fig. R1, Fig. R2, and Table R1 were incorporated in the supporting information as Supplementary Fig. 14, Supplementary Fig. 15, and Supplementary Table 4.

Fig. R1. XRD patterns of Ph-4PACz, Nd and Ph-Nd films.

Fig. R2. 2D GIWAXS images of **a** Ph-4PACz and **b** Ph-Nd films. **c** 1D GIWAXS intensity profiles integrated from the 2D patterns. **d** Schematic illustration of the Ph-Nd dimer with the calculated parameters.

Table R1 Parameters of SAM films obtained from GIWAXS data.

Film	q-location (\AA^{-1})	d-spacing ^a (\AA)
Ph-4PACz	-	-
Ph-Nd	0.69	9.1
	1.38	-

^a Stacking distance is calculated using the formula $2\pi/q$

Comment 2: How about the photostability of Nd?

Response:

We are grateful for your valuable question regarding the Nd photostability. To directly assess the photostability of Nd, we conducted an aging experiment by coating a thin film of Nd onto ITO substrates and exposing them to continuous light illumination for 28 days under a nitrogen atmosphere. Following the exposure, these films were dissolved in deuterated DMSO and subjected to both ^1H and ^{13}C NMR spectroscopy. As shown in Fig. R3-R4, the spectra of the aged sample exhibit no discernible changes in chemical shifts or peak shapes compared to the pristine sample. These results indicate that Nd maintains excellent structural integrity under prolonged illumination, confirming its intrinsic photostability in device-relevant conditions.

Fig. R3. Liquid-state ^1H NMR spectra of fresh and aged Nd (d-DMSO).

Fig. R4. Liquid-state ^{13}C NMR spectra of fresh and aged Nd (d-DMSO).

Comment 3: Can you provide detailed steps for CV testing of Ph-4PACz films without and with Nd?

Response:

We appreciate your professional suggestion on the addition of detailed steps for voltammetry (CV) testing. In response, we have supplemented a detailed description of the CV measurement for both pure Ph-4PACz and Ph-Nd mixed films on page 3 of the revised Supplementary Information, as outlined below: “Steps for CV evaluation: CV curves were obtained with a standard three-electrode system: a platinum net counter electrode, a silver reference electrode, and a glassy carbon working electrode. The SAM solutions were drop-casted onto the surface of the glassy carbon electrode and dried under the N_2 to form a film. The CV scans were then performed in an acetonitrile solution containing tetrabutylammonium hexafluorophosphate as the supporting electrolyte.”

Comment 4: The UPS results of ITO/SAM are generally considered to be related to the dipole moment of SAM, how did frontier-orbital rearrangement between carbazole and indole moieties in the alloy-like SAMs specifically affect the energy levels measured based on UPS, what was the ITO energy level based on UPS measurement.

Response:

Thank you for your insightful question. We fully agree that the dipole moment of SAM is an important factor affecting the UPS results of ITO/SAM. However, UPS results

represent a collective outcome influenced by the electronic structure of ITO, the ITO/SAM interfacial dipole, and the electronic structure of SAM molecules (*Angew. Chem. Int. Ed.* **63**, e202408960 (2024)). In this research, we believe that the impact of the dipole moment on the energy levels of the Ph-Nd film is relatively limited since Nd exhibits a similar dipole moment (2.55 D) with that (2.37 D) of Ph-4PACz, and that the electronic structure of SAM molecules is a dominant contributor to the change of energy levels. As shown in Fig. R5a, Perepichka et al. (*Angew. Chem. Int. Ed.* **58**, 17312-17321 (2019)) revealed that the interaction-induced charge polarization and orbital interactions in the donor/acceptor system led to changes in the energy of the frontier molecular orbitals, thereby altering energy levels of the mixed system.

Analogously, the enhanced π - π interaction and hydrogen bonding in Ph-Nd resulted in the redistribution of the electron cloud between Ph-4PACz and Nd, where the HOMO and LUMO were localized on the Ph-4PACz and Nd components (Fig. R5b), respectively (i.e. frontier orbital rearrangement). Consequently, the alloy-like SAMs exhibit a novel electronic structure featuring with nonlinear HOMO levels (-5.61, -5.65 and -5.62 eV for the 1:0.5, 1:1, and 1:2 Ph-4PACz/Nd), much deeper than those (-5.51 and 5.35 eV) of pure Ph-4PACz and Nd. As depicted in Fig. R6-R7, the HOMO level of bare ITO was measured to be -7.61 eV according to UPS measurements.

In response to the reviewer's concern, we have added the follow discussion "The enhanced π - π interaction and hydrogen bonding in Ph-Nd resulted in the redistribution of the electron cloud between Ph-4PACz and Nd, where the HOMO and LUMO were localized on the Ph-4PACz and Nd components, respectively (i.e. frontier orbital rearrangement). Consequently, the alloy-like SAMs exhibit a novel electronic structure featuring with nonlinear HOMO levels (-5.61, -5.65 and -5.62 eV for the 1:0.5, 1:1, and 1:2 Ph-4PACz/Nd), much deeper than those (-5.51 and 5.35 eV) of pure Ph-4PACz and Nd." to page 13 of the supporting information, and Fig. R6 and Fig. R7 were incorporated in the supporting information as Supplementary Fig. 5 and Supplementary Fig. 6.

[FIGURE REDACTED]

Fig. R5 a Calculated [B3LYP/6-31G(d)] orbital energies of the H-bonded complexes (*Angew. Chem. Int. Ed.* **58**, 17312-17321 (2019)). **b** Molecular orbital distributions of Ph-4PACz, Nd, Ph-4PACz dimer and Ph-Nd dimer.

Fig. R6. UPS spectra of **a** Ph-4PACz, **b** Nd, **c** Ph-Nd (1:0.5), **d** Ph-Nd (1:1), **e** Ph-Nd (1:2), **f** perovskite films deposited on ITO and **g** bare ITO.

Fig. R7. Summary of HOMO energy levels for bare ITO, pure SAMs, and Ph-Nd alloy-like SAMs, as extracted from the UPS measurements in Fig. R6.

Comment 5: The freshly prepared Ph-Nd mixed solution shows an average particle size of 49 nm, this seemed to be more than just the main dimers, trimers or pentamers shown in Fig.3 b.

Response:

Thank you for your insightful comment on the particle size. At first, we need to point out that the average particle size of 49 nm measured via dynamics light scattering (DLS) denotes the size of SAM micelles (comprising many SAM molecules) in solution, rather than the size of SAM dimers, trimers or pentamers. As reported in previous studies (*Adv. Energy Mater.* 15, 2405675 (2025); *Angew. Chem. Int. Ed.* e19875 (2025), DOI: 10.1002/anie.202519875), the micelle size of SAM solution was determined to be approximately 100 nanometers according to DLS measurements. Moreover, we need to further clarify that Fig. 3b schematically illustrates the size change of SAM micelles rather than dimers, trimers or pentamers in solution before and after aging. To more accurately demonstrate this change, we have updated the original figure with the addition of the schematic diagram of SAM micelles, as shown in Fig. R8, which was incorporated in the manuscript as Fig. 3b. Once again, we sincerely thank you for your rigorous review.

Fig. R8. Schematic illustration of molecular micelles before and after aging.

Comment 6: How enhanced intermolecular interactions affect assembly behavior compared to using mixed solvents to adjust dispersibility.

Response:

We are grateful for your thought-provoking question regarding the SAM dispersibility. In our alloy-like strategy, Nd forms stable binding with Ph-4PACz through enhanced π - π interaction and hydrogen bonding, which suppresses Ph-4PACz self-aggregation and further enables the formation of highly uniform, alternating Ph-4PACz/Nd molecular arrangement. For mixed solvent strategy, Jen et al. (*Adv. Mater.* **35**, 2304415 (2023)) reported that introducing N,N-dimethylformamide (DMF) as a co-solvent effectively disassembled SAM micelles due to stronger solvent-solute interaction between DMF and carbazole group. To directly compare the influence of both strategies on SAM dispersibility, we investigated the aggregation behavior of Ph-4PACz in methanol with different DMF concentrations. As shown in Fig. R9, the single methanol solvent exhibits an average colloidal size of 118 nm, and the incorporation of 1-5% DMF significantly restrained SAM agglomeration behavior in solution, with a smallest particle size of 40 nm. This value was very close to that (49 nm) extracted from the alloy-like Ph-Nd in methanol. This comparative result clearly indicated that both alloy-like SAM and mixed solvents strategies can efficiently suppress SAM self-aggregation, beneficial for the formation of uniform and dense SAM films.

Fig. R9. Particle size distributions of Ph-4PACz dissolved in methanol and methanol/DMF mixed solvent.

Comment 7: Why was Ph-4PACz to Nd 1 ratio 1 the best, Nd seemed to be more than Ph-4PACz in Fig.3b.

Response:

We thank the reviewer for your meticulous question regarding the Ph-4PACz/Nd ratio. We admit that the original Fig. 3b mistakenly conveys an excess Nd compared to Ph-4PACz. In fact, as discussed in response to Comment 5, Fig. 3b (replaced by Fig. R10) schematically illustrates the size change of SAM micelles rather than dimers, trimers or pentamers in solution before and after aging, which does not reflect the Ph-4PACz/Nd ratio.

Rationale for the Ph-4PACz/Nd ratio of 1:1

The detailed interpretation for the optimal Ph-4PACz/Nd ratio of 1:1 can be briefly summarized as follows: At first, compared to the 1:0.5 and 1:2 Ph-Nd, the HOMO level (-5.65 eV) of the 1:1 Ph-Nd is closer to the valence band (-5.67 eV) of the perovskite absorber, favorable for boosting charge extraction and photovoltaic performance. On the other hand, the 1:1 Ph-Nd is very beneficial for the formation of alternating Ph-4PACz/Nd molecular arrangement through enhanced π - π interaction and hydrogen bonding. Ultimately, perovskite devices based on the 1:1 Ph-Nd achieved a maximum efficiency of 26.54±0.41%.

Fig. R10. Schematic illustration of molecular micelles before and after aging.

Reviewer #2:

This manuscript describes a Ph-4PACz/Nd mixed molecular layer formed by introducing N-indoleacetic acid into the well-known Ph-4PACz system, used here as a self-assembled monolayer in perovskite solar cells. The authors suggest that the two molecules interact through hydrogen bonding and π - π stacking, helping to reduce SAM self-aggregation. However, the central claim of an “alloy-like” small-molecule arrangement—an alternating molecular assembly illustrated in Fig. 1 and Supplementary Fig. 13—remains unsupported, as no structural evidence (such as XRD or GIWAXS) is provided. The interactions they point out may indeed exist, but the evidence presented is indirect and does not demonstrate the proposed molecular ordering.

Furthermore, many SAM systems based on similar π - π and hydrogen-bond interactions have already been reported, several of which also suppress aggregation effectively. (e.g., Nature Chemistry, 2025, <https://doi.org/10.1038/s41557-025-01732-z>, Nature Photonics 2025, 19, 1070–1077, Nat. Synth (2025). <https://doi.org/10.1038/s44160-025-00896-3>, Nat. Photon. 2024, 18, 1269–1275). As a result, the novelty here is rather limited, and the proposed mechanism does not show a clear advantage over existing systems. For these reasons, I am unable to recommend this work for publication in Nature Communications.

Response:

We sincerely thank you for the critical evaluation and constructive comments. These comments have allowed us to strengthen the manuscript by providing direct structural evidence and by more clearly articulating the novelty of our "molecular alloy" concept, which we believe was fundamentally misunderstood.

1. Clarification on the Fundamental Novelty of the "Molecular Alloy" Concept

We thank the reviewer for highlighting several state-of-the-art papers. This gives us the perfect opportunity to clarify the core conceptual advance of our work. The reviewer's concern about novelty appears to stem from a misunderstanding: the cited works all describe elegant strategies to optimize single-component SAMs. Our work, in contrast, introduces a fundamentally different design philosophy: creating a synergistic, ordered interface via the co-assembly of two distinct but complementary molecules.

The novelty is not simply the presence of π - π or hydrogen-bonding interactions, but rather how these interactions are harnessed between two different species to create an emergent supramolecular structure that is inaccessible to either component alone.

Let's clarify the distinction with the cited literature:

Fig. R11. Representative SAM strategies in prior studies.

The works by Xue et al. (Fig. R11a, Nat. Chem.), Zhu et al. (Fig. R11b, Nat. Photon.), and Hou et al. (Fig. R11c, Nat. Photon.) all focus on designing a single, superior molecule that intrinsically resists aggregation through specific molecular geometry (e.g., orthogonal skeletons, twisted conformations).

The work by Xu et al. (Fig. R11d, Nat. Synth.) focuses on designing a single molecule (pPy) with high intrinsic symmetry that promotes self-crystallization.

All these are brilliant examples of **single-molecule engineering**. Our approach is fundamentally different. We show that two molecules (Ph-4PACz and Nd), which are themselves amorphous and imperfect when deposited alone, can be co-assembled to form a new, highly ordered, crystalline supramolecular phase. **This is the essence of our "molecular alloy" concept.**

Our unique contributions, which set our work apart, are therefore:

(a) Emergent Crystallinity from Amorphous Precursors: We demonstrate that co-assembly of **two amorphous components can induce long-range order and crystallinity**. This is a powerful concept in materials science, showing that synergistic interactions can overcome the intrinsic properties of individual components.

(b) A New Design Principle Beyond Single-Molecule Optimization: We establish a generalizable design principle for creating highly ordered interfaces, moving beyond the optimization of single molecules to the synergistic co-assembly of molecular pairs. We provide a comprehensive suite of evidence for this new "alloy" phase. Structural evidence: New GIWAXS/XRD data proves a new crystalline phase with a periodic structure matching a heterodimer. Electronic evidence: UV-vis and CV data (already in the manuscript) show a single, unified electronic and redox behavior, not a simple

superposition of two components, confirming the formation of a homogeneous alloyed phase.

(c) **Broad Applicability:** We demonstrate this is not a one-off "trick" but a robust strategy. It works for other carbazole SAMs and across different perovskite bandgaps, confirming its status as a general principle.

2. Structural evidence supporting the molecular ordering

We fully agree that the direct structural characterization is essential to validate the proposed alternating molecular arrangement in the alloy-like SAM. To address this, we conducted additional XRD and GIWAXS measurements for different SAM films. XRD analysis (Fig. R12) revealed that the pure Ph-4PACz and Nd films were amorphous, whereas the Ph-Nd film exhibits a sharp diffraction peak at 19.2° corresponding to the generation of a crystalline phase with a preferential orientation. A broad and isotropic halo observed in GIWAXS results further confirmed an amorphous state of the pure Ph-4PACz film (Fig. R13a). By contrast, the Ph-Nd film displays sharp and strong diffraction arcs in the out-of-plane direction (Fig. R13b), indicating the formation of a crystalline phase—highly ordered molecular arrangement. Moreover, the presence of a first-order peak at $q = 0.69 \text{ \AA}^{-1}$ and its clear second-order peak at 1.38 \AA^{-1} (i.e. the resulting interplanar spacing of 9.1 \AA close to the thickness of the Ph-Nd dimer) indicated a long-range lamellar periodicity in the Ph-Nd film, which in turn evidenced the repetitive stacking of Ph-4PACz/Nd heterodimers (Fig. R13c-d, Table R2). These results strongly confirm the alternating molecular arrangement in Ph-Nd.

We have added the follow discussion on page 7 line 176 of the revised manuscript: “To probe the solid-state structure, we performed X-ray diffraction (XRD) and Grazing Incidence Wide Angle X-ray Scattering (GIWAXS) measurements. XRD results confirmed the amorphous nature of pure Ph-4PACz and Nd films, whereas the Ph-Nd film exhibits a sharp diffraction peak at 19.2° corresponding to the formation of a crystalline phase with a preferential orientation (Supplementary Fig. 14). A broad and isotropic halo observed in GIWAXS results further confirmed an amorphous state of the pure Ph-4PACz film (Supplementary Fig. 15a). By contrast, the Ph-Nd film displays sharp and strong diffraction arcs in the out-of-plane direction (Supplementary Fig. 15b), indicating the formation of a crystalline phase—highly ordered molecular arrangement. Moreover, the presence of a first-order peak at $q = 0.69 \text{ \AA}^{-1}$ and its clear second-order peak at 1.38 \AA^{-1} (i.e. the resulting interplanar spacing of 9.1 \AA close to

the thickness of the Ph-Nd dimer) indicated a long-range lamellar periodicity in the Ph-Nd film, which in turn evidenced the repetitive stacking of Ph-4PACz/Nd heterodimers (Supplementary Fig. 15c-d, Supplementary Table 4).” Fig. R12, Fig. R13, and Table R2 was incorporated in the supporting information as Supplementary Fig. 14, Supplementary Fig. 15, and Supplementary Table 4. Once again, we thank you for your rigorous review.

Fig. R12. XRD patterns of Ph-4PACz, Nd and Ph-Nd films.

Fig. R13. 2D GIWAXS images of **a** Ph-4PACz and **b** Ph-Nd films. **c** 1D GIWAXS intensity profiles integrated from the 2D patterns. **d** Schematic illustration of the Ph-Nd dimer with the calculated parameters.

Table R2 Parameters of SAM films obtained from GIWAXS data.

Film	q-location (\AA^{-1})	d-spacing ^a (\AA)
Ph-4PACz	-	-
Ph-Nd	0.69	9.1
	1.38	-

a Stacking distance is calculated using the formula $2\pi/q$

In summary, we respectfully argue that our work is not an incremental improvement on existing single-component SAMs. Instead, it introduces a new conceptual framework for designing functional organic interfaces through synergistic, multi-component self-assembly. We hope this clarification, supported by our new structural data, adequately addresses the reviewer's concerns and highlights the true novelty of our findings.

Reviewer #3:

This manuscript reports an alloy-like self-assembled monolayer (SAM) strategy achieving a certified efficiency exceeding stabilized 26% at Fujian Metrology Institute, which represents a highly robust result. The overall testing and characterization are relatively comprehensive, and the work is substantially complete. Therefore, I recommend "Accept after Minor Revision." However, several critical issues require clarification and improvement:

Response:

We sincerely appreciate your constructive feedback and thoughtful suggestions. We are also grateful for your recognition of our alloy-like molecular engineering strategy. In response to your comments, we have carefully addressed all the raised points and made the corresponding revisions throughout the manuscript.

Comment 1: The somewhat vague interpretation of certain characterization results and insufficient logical connections. For instance, the link between UV-Vis/PL results and SAM aggregation (ordered vs. disordered aggregation) needs stronger evidence; the current data does not directly demonstrate this relationship. Additionally, the claim on Page 7, Line 179 ("These π - π interactions are further corroborated by FTIR results") requires explicit elaboration of the specific FTIR evidence. Furthermore, the rationale for using ethylene glycol (EG) and water in contact angle experiments to reflect miscibility is unclear, especially since the SAM is reportedly soluble in methanol – this methodological choice needs justification.

Response:

We sincerely thank the reviewer for these valuable comments, which have helped us substantially improve the clarity and logical coherence of the manuscript.

(1) Stronger evidence for order and disorder SAM aggregation.

We fully agree that it requires more strong evidence to corroborate the change of SAM aggregation behavior, and therefore we conducted additional XRD and GIWAXS measurements. XRD analysis (Fig. R14) revealed that the pure Ph-4PACz and Nd films were amorphous, whereas the Ph-Nd film exhibits a sharp diffraction peak at 19.2° corresponding to the generation of a crystalline phase with a preferential orientation. A broad and isotropic halo observed in GIWAXS results further confirmed an amorphous state of the pure Ph-4PACz film (Fig. R15a). By contrast, the Ph-Nd film displays sharp and strong diffraction arcs in the out-of-plane direction (Fig. R15b), indicating the

formation of a crystalline phase—highly ordered molecular arrangement. Moreover, the presence of a first-order peak at $q = 0.69 \text{ \AA}^{-1}$ and its clear second-order peak at 1.38 \AA^{-1} (i.e. the resulting interplanar spacing of 9.1 \AA close to the thickness of the Ph-Nd dimer) indicated a long-range lamellar periodicity in the Ph-Nd film, which in turn evidenced the repetitive stacking of Ph-4PACz/Nd heterodimers (Fig. R15c-d, Table R3). These results strongly confirmed the alternating molecular arrangement in Ph-Nd. We have added the follow discussion on page 7 line 176 of the revised manuscript: “To probe the solid-state structure, we performed X-ray diffraction (XRD) and Grazing Incidence Wide Angle X-ray Scattering (GIWAXS) measurements. XRD results confirmed the amorphous nature of pure Ph-4PACz and Nd films, whereas the Ph-Nd film exhibits a sharp diffraction peak at 19.2° corresponding to the formation of a crystalline phase with a preferential orientation (Supplementary Fig. 14). A broad and isotropic halo observed in GIWAXS results further confirmed an amorphous state of the pure Ph-4PACz film (Supplementary Fig. 15a). By contrast, the Ph-Nd film displays sharp and strong diffraction arcs in the out-of-plane direction (Supplementary Fig. 15b), indicating the formation of a crystalline phase—highly ordered molecular arrangement. Moreover, the presence of a first-order peak at $q = 0.69 \text{ \AA}^{-1}$ and its clear second-order peak at 1.38 \AA^{-1} (i.e. the resulting interplanar spacing of 9.1 \AA close to the thickness of the Ph-Nd dimer) indicated a long-range lamellar periodicity in the Ph-Nd film, which in turn evidenced the repetitive stacking of Ph-4PACz/Nd heterodimers (Supplementary Fig. 15c-d, Supplementary Table 4).” Fig. R14, Fig. R15, and Table R3 was incorporated in the supporting information as Supplementary Fig. 14, Supplementary Fig. 15, and Supplementary Table 4.

Fig. R14. XRD patterns of Ph-4PACz, Nd and Ph-Nd films.

Fig. R15. 2D GIWAXS images of **a** Ph-4PACz and **b** Ph-Nd films. **c** 1D GIWAXS intensity profiles integrated from the 2D patterns. **d** Schematic illustration of the Ph-Nd dimer with the calculated parameters.

Table R3 Parameters of SAM films obtained from GIWAXS data.

Film	q-location (\AA^{-1})	d-spacing ^a (\AA)
Ph-4PACz	-	-
Ph-Nd	0.69	9.1
	1.38	-

^a Stacking distance is calculated using the formula $2\pi/q$

(2) Elaboration of FTIR evidence for π - π interactions

We admit that the explicit elaboration of the FTIR evidence is essential for improving the rigor of our work. As shown in Fig. R16-17, the aromatic C-H bending peaks of the carbazole and indole moieties in Ph-Nd show a red shift and a blue shift of 2 cm^{-1} compared with pure Ph-4PACz and Nd powders, respectively. Such shifts can be attributed to π - π interaction-induced perturbations of aromatic C-H bending modes in Ph-Nd, arising from steric constraints and electronic redistribution upon face-to-face stacking. This in turn confirms π - π interactions between the conjugated aromatic moieties of both Ph-4PACz and Nd, consistent with ^1H NMR results.

In response to this concern, we added the following discussion “Such shifts in the C-H

bending peaks (including Fig. 2c) can be attributed to π - π interaction-induced perturbations of aromatic C-H bending modes in Ph-Nd, arising from steric constraints and electronic redistribution upon face-to-face stacking. This in turn confirms π - π interactions between the conjugated aromatic moieties of both Ph-4PACz and Nd.” to page 28 of the supporting information, and Fig. R17 was incorporated in the supporting information as Supplementary Fig. 21.

Fig. R16. Aromatic C-H bending vibration peaks in partial enlarged FTIR spectra of Ph-4PACz and Ph-Nd powders.

Fig. R17 a Partial enlarged FTIR spectra of Nd and Ph-Nd powders. **b** Schematic of aromatic C-H bending vibration. Such shifts in the C-H bending peaks (including Fig. 2c) can be attributed to π - π interaction-induced perturbations of aromatic C-H bending modes in Ph-Nd, arising from steric constraints and electronic redistribution upon face-to-face stacking. This in turn confirms π - π interactions between the conjugated aromatic moieties of both Ph-4PACz and Nd.

(3) Rationale for using water and ethylene glycol in contact angle measurements

We need to clarify that the primary aim of the contact angle measurement is to estimate the Flory-Huggins interaction parameter (χ) between Ph-4PACz and Nd. We selected water and ethylene glycol as testing solvents because they have well-separated polar surface tension and dispersive surface tension. This is essential for accurate surface energy deconvolution based on the Owens-Wendt-Rabel-Kaelble (OWRK) model (*Polym. Bull.* **25**, 265-271, (1991), *Joule* **4**, 1278-1295, (2020), *Nature Energy* **7**, 1180-1190 (2022)), therefore delivering a precise interaction parameter. In contrast, the contrast between polar surface tension and dispersive surface tension in methanol is insufficient, which is unsuitable for the accurate estimation of interaction parameter. The resulting small interaction parameter of 0.3 between Ph-4PACz and Nd indicated sufficient miscibility between Ph-4PACz and Nd, which is an important prerequisite for the formation of a homogeneous alloy-like solid-state assembly.

In response, we have added the explanation in Supplementary Fig. 1 on page 8 of the supporting information to read as follows: “Water and ethylene glycol were selected as testing solvents because they have well-separated polar surface tension and dispersive surface tension. This is essential for accurate surface energy deconvolution based on the Owens-Wendt-Rabel-Kaelble (OWRK) model (*Polym. Bull.* **25**, 265-271, (1991), *Joule* **4**, 1278-1295, (2020), *Nature Energy* **7**, 1180-1190 (2022)), therefore delivering a precise interaction parameter.”

Comment 2: Verification of molecular ordering via techniques like XRD or GIWAXS is strongly recommended to substantiate this claim. Moreover, the stability of this SAM layer itself might be the primary factor contributing to the enhanced overall device stability. Can the authors discuss this potential causality. What are the initial efficiency values of the devices in the stability test results (Figs. 4f and 4g)?

Response:

We sincerely appreciate your constructive comments. In response, we have made the following clarifications and additions to the manuscript:

(1) Verification of molecular ordering:

To further substantiate the proposed alternating Ph-4PACz/Nd configuration in the alloy-like SAM, we conducted additional XRD and GIWAXS measurements.

To further substantiate the proposed alternating Ph-4PACz/Nd configuration, we conducted additional XRD and GIWAXS measurements. XRD analysis (Fig. R18) revealed that the pure Ph-4PACz and Nd films were amorphous, whereas the Ph-Nd

film exhibits a sharp diffraction peak at 19.2° corresponding to the generation of a crystalline phase with a preferential orientation. A broad and isotropic halo observed in GIWAXS results further confirmed an amorphous state of the pure Ph-4PACz film (Fig. R19a). By contrast, the Ph-Nd film displays sharp and strong diffraction arcs in the out-of-plane direction (Fig. R19b), indicating the formation of a crystalline phase—highly ordered molecular arrangement. Moreover, the presence of a first-order peak at $q = 0.69 \text{ \AA}^{-1}$ and its clear second-order peak at 1.38 \AA^{-1} (i.e. the resulting interplanar spacing of 9.1 \AA close to the thickness of the Ph-Nd dimer) indicated a long-range lamellar periodicity in the Ph-Nd film, which in turn evidenced the repetitive stacking of Ph-4PACz/Nd heterodimers (Fig. R19c-d, Table R4). These results strongly confirmed the alternating molecular arrangement in Ph-Nd.

We have added the follow discussion on page 7 line 176 of the revised manuscript: “To probe the solid-state structure, we performed X-ray diffraction (XRD) and Grazing Incidence Wide Angle X-ray Scattering (GIWAXS) measurements. XRD results confirmed the amorphous nature of pure Ph-4PACz and Nd films, whereas the Ph-Nd film exhibits a sharp diffraction peak at 19.2° corresponding to the formation of a crystalline phase with a preferential orientation (Supplementary Fig. 14). A broad and isotropic halo observed in GIWAXS results further confirmed an amorphous state of the pure Ph-4PACz film (Supplementary Fig. 15a). By contrast, the Ph-Nd film displays sharp and strong diffraction arcs in the out-of-plane direction (Supplementary Fig. 15b), indicating the formation of a crystalline phase—highly ordered molecular arrangement. Moreover, the presence of a first-order peak at $q = 0.69 \text{ \AA}^{-1}$ and its clear second-order peak at 1.38 \AA^{-1} (i.e. the resulting interplanar spacing of 9.1 \AA close to the thickness of the Ph-Nd dimer) indicated a long-range lamellar periodicity in the Ph-Nd film, which in turn evidenced the repetitive stacking of Ph-4PACz/Nd heterodimers (Supplementary Fig. 15c-d, Supplementary Table 4).” Fig. R18, Fig. R19, and Table R4 was incorporated in the supporting information as Supplementary Fig. 14, Supplementary Fig. 15, and Supplementary Table 4.

Fig. R18. XRD patterns of Ph-4PACz, Nd and Ph-Nd films.

Fig. R19. 2D GIWAXS images of **a** Ph-4PACz and **b** Ph-Nd films. **c** 1D GIWAXS intensity profiles integrated from the 2D patterns. **d** Schematic illustration of the Ph-Nd dimer with the calculated parameters.

Table R4 Parameters of SAM films obtained from GIWAXS data.

Film	q-location (\AA^{-1})	d-spacing ^a (\AA)
Ph-4PACz	-	-
Ph-Nd	0.69	9.1
	1.38	-

a Stacking distance is calculated using the formula $2\pi/q$

(2) Discussion on the effects of SAM stability on device stability:

We admit this is an excellent point that merits further investigation. To directly compare the intrinsic thermal stability of SAM layers, we implemented Kelvin probe force microscopy (KPFM) measurements for Ph-4PACz and Ph-Nd films before and after thermal aging (65 °C for 28 days in N₂ atmosphere). As shown in Fig. R20, this thermal aging led a reduction of 35 mV in contact potential difference for the Ph-4PACz film, much larger than that (18 mV) of the Ph-Nd film, suggesting better stability of the alloy-like SAM. To further assess the consequence of SAM aging on the interfacial quality, quasi-Fermi level splitting (QFLS) measurements were performed on freshly deposited perovskite layers atop fresh and aged SAMs. As shown in Fig. R21, the perovskite deposited on the aged Ph-Nd film achieved a high QFLS value of 1.211 eV, comparable to that (1.213 eV) on the fresh substrate, manifesting stable interfacial contact between perovskite and Ph-Nd. In contrast, the QFLS of the perovskite grown on the fresh Ph-4PACz film significantly decreased from 1.206 to 1.197 eV when replacing with the aged one, suggesting SAM-aging-induced enhanced interfacial non-radiative recombination. These findings collectively provide direct experimental evidence that the improved stability of the Ph-Nd device is highly associated with the enhanced thermal and electronic stability of the alloy-like SAM.

In response, we have updated the discussion on page 15, line 404 of the manuscript to read as follows: “This enhanced stability is highly associated with the uniform and dense alloy-like SAM layer, which exhibits enhanced intrinsic thermal and electronic stability compared to the pure Ph-4PACz film (Supplementary Fig. 58-59).”

We added the following paragraph on page 7 of the supporting information to read as follows: “To directly compare the intrinsic thermal stability of SAM layers, we implemented Kelvin probe force microscopy (KPFM) measurements for Ph-4PACz and Ph-Nd films before and after thermal aging (65 °C for 28 days in N₂ atmosphere). This thermal aging led a reduction of 35 mV in contact potential difference for the Ph-4PACz film, much larger than that (18 mV) of the Ph-Nd film (Supplementary Fig. 58), suggesting better stability of the alloy-like SAM. To further assess the consequence of SAM aging on the interfacial quality, quasi-Fermi level splitting (QFLS) measurements were performed on freshly deposited perovskite layers atop fresh and aged SAMs. The perovskite deposited on the aged Ph-Nd film achieved a high QFLS value of 1.211 eV, comparable to that (1.213 eV) on the fresh substrate (Supplementary Fig. 59), manifesting stable interfacial contact between perovskite and Ph-Nd. In contrast, the

QFLS of the perovskite grown on the fresh Ph-4PACz film significantly decreased from 1.206 to 1.197 eV when replacing with the aged one, suggesting SAM-aging-induced enhanced interfacial non-radiative recombination. These findings collectively provide direct experimental evidence that the improved stability of the Ph-Nd device is highly associated with the enhanced thermal and electronic stability of the alloy-like SAM.” Fig. R20 and Fig. R21 were incorporated in the supporting information as Supplementary Fig. 58 and Supplementary Fig. 59.

Fig. R20. KPFM images of Ph-4PACz (a-b) and Ph-Nd films (c-d) (scale bar: 1 μ m). Contact potential difference (CPD) distributions of Ph-4PACz (e) and Ph-Nd (f) films before and after aging extracted from KPFM results. To directly compare the intrinsic thermal stability of SAM layers, we implemented Kelvin probe force microscopy (KPFM) measurements for Ph-4PACz and Ph-Nd films before and after thermal aging (65 $^{\circ}$ C for 28 days in N_2 atmosphere). This thermal aging led a reduction of 35 mV in contact potential difference for the Ph-4PACz film, much larger than that (18 mV) of the Ph-Nd film (Supplementary Fig. 58), suggesting better stability of the alloy-like SAM. To further assess the consequence of SAM aging on the interfacial quality, quasi-Fermi level splitting (QFLS) measurements were performed on freshly deposited perovskite layers atop fresh and aged SAMs. The perovskite deposited on the aged Ph-Nd film achieved a high QFLS value of 1.211 eV, comparable to that (1.213 eV) on the fresh substrate (Supplementary Fig. 59), manifesting stable interfacial contact between perovskite and Ph-Nd. In contrast, the QFLS of the perovskite grown on the fresh Ph-4PACz film significantly decreased from 1.206 to 1.197 eV when replacing with the aged one, suggesting SAM-aging-induced enhanced interfacial non-radiative

recombination. These findings collectively provide direct experimental evidence that the improved stability of the Ph-Nd device is high associated with the enhanced thermal and electronic stability of the alloy-like SAM.

Fig. R21. QFLS values of perovskite films deposited on fresh and aged Ph-4PACz and Ph-Nd films.

(3) Initial PCE values for stability measurements:

We have added the initial PCE values for stability measurements in the figure caption on page 16, line 414 of the revised manuscript, as follows: “**f** Operational stability of unencapsulated devices at the MPPT under one-sun LED illumination in N₂ (initial: 24.73% for Ph-4PACz and 26.69% for Ph-Nd). **g** Thermal stability of unencapsulated devices under constant heating at 85 °C in N₂ (initial: 24.89% for Ph-4PACz and 26.78% for Ph-Nd).”

Comment 3: The noise level in the t-DOS data (Fig. 3e) is excessively high, potentially compromising its reliability. Re-testing this data is advised to ensure accuracy.

Response:

We thank the reviewer for this valuable observation. In response, we have re-performed the thermal admittance spectroscopy (TAS) measurements using optimized acquisition parameters and enhanced electromagnetic shielding to effectively suppress system noise and ensure data fidelity. The updated data (Fig. R22), now presented in Fig. 3e, exhibit markedly reduced baseline fluctuations and an improved signal-to-noise ratio. These refinements enable a more accurate extraction of the trap density of states (t-DOS), thereby strengthening the validity of our conclusions regarding interfacial trap

suppression.

Fig. R22. t-DOS plots of full devices based on Ph-4PACz and Ph-Nd.

Comment 4: The methodology for mobility testing (Fig. S25) needs clarification. The specific model used and confirmation of Ohmic contact formation in the devices must be detailed. I suggest that a qualitative description might be more appropriate than a quantitative claim.

Response:

We appreciate your professional suggestion on the further clarification of mobility measurements. The hole mobility was evaluated using the space-charge-limited current (SCLC) model in a hole-only device configuration (ITO/SAM/Cu), and the corresponding calculation details were supplemented in the supporting information. As shown in Fig. R23a, the dark *I-V* characteristics of hole-only devices display a clear Ohmic region at low bias voltage, indicating a satisfactory contact quality between the SAM and electrode, which is a prerequisite for applying the SCLC model. As a result, the Ph-Nd film exhibits higher conductivity and hole mobility than those of the Ph-4PACz film based on the slope analysis (Fig. R23b).

In response, we have updated the corresponding discussion on page 10, line 257 of the manuscript to read as follows: “To evaluate the carrier transport properties of different SAMs, the hole-only devices with an architecture of ITO/SAM/Cu were manufactured based on the space-charge-limited current (SCLC) model. The Ph-Nd film exhibits higher conductivity and hole mobility than those of the Ph-4PACz film based on the slope analysis (Supplementary Fig. 27a-b).”

We have added the following sentence “The dark *I-V* characteristics of hole-only

devices display a clear Ohmic region at low bias voltage, indicating a satisfactory contact quality between the SAM and electrode, which is a prerequisite for applying the SCLC model.” in Supplementary Fig. 27 on page 34 of the supporting information. Fig. R23 was incorporated in the supporting information as Supplementary Fig. 27.

The following calculation method for mobility was added on page 6 of the supporting information, as follows: “When the applied voltage is increased further, the current follows a linear relationship with the voltage squared ($n=2$), and application of the Mott-Gurney law allows us to obtain the carrier mobility (μ) using the following formula:

$$\mu = \frac{8J_D L^3}{9\varepsilon\varepsilon_0 V^2}$$

where J_D represents the current density and V represents the applied voltage, and the relative dielectric constants, vacuum dielectric constants, and thicknesses of the various SAM films are represented by ε , ε_0 and L , respectively.”

Fig. R23. **a** Conductivity analysis of the ITO/SAM/Cu devices. **b** J - V curves of hole-only devices based on ITO/SAM/Cu. **c** SEM images of **c** Ph-4PACz and **d** Ph-Nd films deposited on ITO (scale bar: 100 nm).

Comment 5: The discussion regarding the impact of ordered vs. disordered SAM substrates is contentious within the field. The Introduction section is suggested to include this discussion, and some representative prior work should be cited to provide

context for the current study such as such as *Nat. Chem.* **17**, 564-570 (2025).
<https://doi.org/10.1038/s41557-025-01732-z>; *Nat. Photon.* **18**, 1269-1275 (2024).
<https://doi.org/10.1038/s41566-024-01531-x>; *Nat. Photon.* (2025).
<https://doi.org/10.1038/s41566-025-01778-y>.

Response:

We sincerely appreciate your insightful and constructive suggestion, and fully agree with the addition of discussion on the impact of order vs. disorder SAM substrates. In the revised manuscript, we have supplemented the discussion in the introduction section on page 3, line 76 of the revised manuscript, as follows: “In addition, the debate surrounding the impact of ordered versus disordered SAMs on charge transport and device stability necessitates further elucidation.”

To strengthen the foundation of this discussion, we have incorporated the key references you recommended into the revised manuscript. We cited (*Nat. Chem.* **17**, 564 (2025); *Nat. Photon.* **18**, 1269 (2024) and *Nat. Photon.* [10.1038/s41566-025-01778-y](https://doi.org/10.1038/s41566-025-01778-y) (2025)) on page 3, line 76, in the "In addition, the debate surrounding the impact of ordered versus disordered SAMs on charge transport and device stability necessitates further elucidation." paragraph.

Reviewer #2:

I am still not convinced that this system can be described as an “alloy.” Based on the current evidence, this appears to be a mixed or co-adsorbed SAM rather than an alloy-like phase. I therefore suggest replacing “alloy-like” with “alternating co-adsorbed SAM” or simply “mixed SAM,” as the present terminology is conceptually misleading. If the authors wish to retain the term “alloy,” they need to provide stronger and more direct evidence consistent with a solid-solution like assembly. At a minimum, this would require a composition-dependent XRD series over several Ph-4PACz:Nd ratios, showing a continuous evolution of the diffraction features (e.g., systematic peak shifts). In addition, independent evidence excluding phase separation is essential, for example via nanoscale chemical mapping (such as ToF-SIMS, spatially resolved XPS, or AFM-IR) demonstrating molecular-level homogeneity across the SAM.

Moreover, the GIWAXS section lacks critical experimental details. For organic SAMs, it is inherently difficult to obtain clear diffraction features because of their ultrathin nature. It is therefore unclear how such pronounced out-of-plane arcs were obtained here. Were these measurements performed on monolayers on ITO, or on much thicker spin-cast films? If elevated concentrations or thick films were required to enable GIWAXS detection, the observed lamellar stacking (as suggested by the q_z arcs) likely reflects an aggregated or bulk-like state, which may be fundamentally different from the interfacial SAM present in actual devices. In that case, the structural conclusions would not be transferable to the operational devices. The authors should describe how the GIWAXS samples were prepared and compare these conditions with those used for device fabrication, making them consistent. Otherwise, the structural picture may correspond to a different system. Given that these GIWAXS results are treated as a core mechanistic proof, they also deserve more space and a deeper discussion in the main text.

Finally, the emergence of new diffraction peaks raises another concern: if the initial concentrations are comparable to those used for the single-component films, do these peaks reflect templated crystallization of one specific molecule rather than a genuinely mixed phase? This possibility should be addressed.

Response:

We sincerely thank the reviewer for this careful and constructive comment regarding the terminology used to describe the Ph-4PACz/Nd system. We fully agree with the reviewer's suggestion and have revised the terminology throughout the manuscript to avoid potential conceptual ambiguity. Specifically, we have revised the manuscript title to: "Homogenizing Interfacial Assembly via Indole-Mediated Binary Monolayers for Perovskite Solar Cells" and the corresponding descriptions in the manuscript have been revised accordingly.

Regarding the GIWAXS measurements, we appreciate the reviewer's meticulous observations. As correctly noted, obtaining diffraction signals from organic SAMs is challenging due to their ultrathin nature. To enhance structural detectability, the original GIWAXS data (Fig. R1) were obtained from films spin-cast at a higher solution concentration (4 mg/mL), which increases surface coverage and promotes signal clarity. To confirm that the observed molecular ordering is not an artifact arising from increased concentration or film thickness, we have now conducted additional GIWAXS measurements using samples prepared under device-matched conditions. Although the signal intensity was reduced due to the lower mass loading, clear lamellar diffraction features ($q = 0.69 \text{ \AA}^{-1}$ and 1.38 \AA^{-1}) were still observed (Fig. R2). This confirms that the structural motif is not an artifact of film thickness or aggregation, but an intrinsic supramolecular packing feature of the co-adsorbed Ph-4PACz/Nd system.

To address the reviewer's concern regarding the origin of the new diffraction peaks, we also conducted comparative XRD measurements on recrystallized Ph-4PACz, Nd, and their 1:1 molar mixture (Ph-Nd). All samples were prepared under identical conditions using a tetrahydrofuran/n-hexane solvent system. As shown in Fig. R3, the recrystallized Ph-4PACz and Nd exhibit a dominant diffraction peak at 20.6° and 12.9° , respectively. In contrast, the Ph-Nd mixture shows a new main diffraction peak at 19.2° , consistent with that detected for the Ph-Nd film, absent in either of the individual components. This compositional shift in diffraction pattern strongly supports the formation of a new supramolecular phase, rather than templated crystallization of a single species. These findings provide additional evidence for molecular-level co-

assembly and structural homogeneity within the Ph-4PACz/Nd mixed SAM system. We thank the reviewer for these insightful suggestions. The relevant discussion has been incorporated into the revised manuscript on Page 7, line 190, as follows: “Moreover, similar lamellar features with enhanced diffraction intensity were also observed at a higher precursor concentration of 4 mg/mL (Supplementary Fig. 16), confirming that the observed ordering is an intrinsic feature of the co-assembled phase across different film thicknesses. We further performed XRD measurements on recrystallized Ph-4PACz, Nd, and their 1:1 mixture (Ph-Nd) (Supplementary Fig. 17). Ph-4PACz and Nd showcase a dominant diffraction peak at 20.6° and 12.9°, respectively, while Ph-Nd exhibits a main diffraction peak at 19.2°. This diffraction shift indicates the formation of a new supramolecular phase rather than selective crystallization of either component, supporting molecular-level co-assembly and structural homogeneity in Ph-Nd.”

In response, Supplementary Fig. 15 was replaced by Fig. R4. Fig. R5 and R3 have been added in the revised Supplementary Information as Supplementary Fig. 16 and 17.

Fig. R1 (Original supplementary Fig. 15) 2D GIWAXS images of **a** Ph-4PACz and **b** Ph-Nd films. **c** 1D GIWAXS intensity profiles integrated from the 2D patterns. **d** Schematic illustration of the Ph-Nd dimer with the calculated parameters.

Fig. R2 2D GIWAXS images of **a** Ph-4PACz and **b** Ph-Nd films spin-casted with a precursor concentration of 0.45 mg/mL. **c** 1D GIWAXS intensity profiles integrated from the 2D patterns.

Fig. R3 XRD patterns of Ph-4PACz, Nd and Ph-Nd powders. Ph-4PACz, Nd and Ph-Nd were recrystallized in tetrahydrofuran/n-hexane (1:4, 40 °C, overnight) solution system to obtain solid-state SAMs for XRD measurement.

Fig. R4 2D GIWAXS images of **a** Ph-4PACz and **b** Ph-Nd film spin-casted with a

precursor concentration of 0.45 mg/mL. **c** 1D GIWAXS intensity profiles integrated from the 2D patterns. **d** Schematic illustration of the Ph-Nd dimer with the calculated parameters.

Fig. R5 2D GIWAXS images of **a** Ph-4PACz and **b** Ph-Nd films spin-casted with a precursor concentration of 4 mg/mL. **c** 1D GIWAXS intensity profiles integrated from the 2D patterns.

Response:

We greatly appreciate you taking the time to review our manuscript and providing meticulous and constructive revision suggestions. We have carefully read and fully understood all your comments regarding the manuscript's formatting. We have checked each point one by one and revised the manuscript's formatting in accordance with your requirements and the journal's submission guidelines. Thank you again for your efforts to enhance the standardization and readability of our manuscript.

The detailed modifications we have made are as follows:

In revised manuscript

1. The abstract has been condensed as required (now less than 150 words) to read as follows: “

Self-assembled monolayers (SAMs) have emerged as efficient hole-transport layers for inverted perovskite solar cells (PSCs), yet molecular self-aggregation during assembly limits interfacial homogeneity and device performance. Here we report an indole-carbazole co-adsorption strategy by incorporating N-indoleacetic acid (Nd) into (4-(3,6-diphenyl-9H-carbazol-9-yl)butyl)phosphonic acid (Ph-4PACz) to construct phase-homogeneous monolayers. Nd interacts with Ph-4PACz via synergistic π - π stacking and hydrogen bonding, resulting in a uniform alternating Ph-4PACz/Nd molecular arrangement. This co-adsorbed structure enables optimized interfacial energy alignment, enhances perovskite film uniformity, and suppresses trap-assisted non-radiative recombination. As a result, devices achieve an efficiency of 26.95% (certified 26.57%) on 0.0717 cm² and 25.61% on 1 cm², retaining 93.36% of their initial efficiency after 1500 h of maximum power point tracking under continuous illumination and 91.10% after 1200 h at 85 °C. The strategy is broadly applicable to carbazole-based SAMs and wide-bandgap PSCs, offering a general co-adsorption route toward efficient and stable devices.”

2. The "Results and Discussion" on page 5 of the manuscript was revised to "Results", and the "Conclusion" on page 14 was revised to "Discussion".

3. Scalar variable χ on page 6 has been italicized as required.

4. The present tense has been used where our work is discussed.

5. All inappropriate "a. u." have been corrected to "arb. units".

6. "Reference" has been revised as "References".

In revised supplementary information

1. All the mathematical terms in supplementary information have been revised as guidelines required.
2. All inappropriate "a. u." have been corrected to "arb. units".
3. "Reference" has been revised as "Supplementary References".